# Stochastic variance-reduced Gaussian variational inference on the Bures–Wasserstein manifold

**Hoang Phuc Hau Luu, Hanlin Yu, Bernardo Williams, Marcelo Hartmann, Arto Klami**
Department of Computer Science
University of Helsinki, Helsinki, Finland
`{hoang-phuc-hau.luu,hanlin.yu,bernardo.williamsmoreno,`
`marcelo.hartmann,arto.klami}@helsinki.fi`

## Abstract

Optimization in the Bures–Wasserstein space has been gaining popularity in the machine learning community since it draws connections between variational inference and Wasserstein gradient flows. The variational inference objective function of Kullback–Leibler divergence can be written as the sum of the negative entropy and the potential energy, making forward-backward Euler the method of choice. Notably, the backward step admits a closed-form solution in this case, facilitating the practicality of the scheme. However, the forward step is not exact since the Bures–Wasserstein gradient of the potential energy involves "intractable" expectations. Recent approaches propose using the Monte Carlo method – in practice a single-sample estimator – to approximate these terms, resulting in high variance and poor performance. We propose a novel variance-reduced estimator based on the principle of control variates. We theoretically show that this estimator has a smaller variance than the Monte-Carlo estimator in scenarios of interest. We also prove that variance reduction helps improve the optimization bounds of the current analysis. We empirically demonstrate that the proposed estimator gains order-of-magnitude improvements over previous Bures–Wasserstein methods.

## 1 Introduction

Variational inference (VI) (Wainwright et al., 2008; Blei et al., 2017) provides a fast and scalable alternative to Markov chain Monte Carlo (MCMC), especially for inference tasks in high dimensions. The main principle of VI is to approximate a complicated distribution $\pi$, e.g., posterior distribution in Bayesian inference, by a simpler tractable family of distributions. The approximation $\mu$ within the family is obtained by solving an optimization problem, providing a closed-form representation and e.g. efficient sampling by construction. The choice of the optimization method is heavily influenced by the assumptions made on the approximation family and the information about $\pi$ that can be obtained, ranging from classical coordinate ascent algorithms for mean-field approximations for targets with conditional conjugacy structure (Blei et al., 2017) to gradient methods using score-function approximations to avoid assumptions on the target density (Ranganath et al., 2014) or flexible approximations parameterized with neural networks (Rezende & Mohamed, 2015).

We focus on Gaussian approximations (Honkela & Valpola, 2004; Opper & Archambeau, 2009; Xu & Campbell, 2022; Quiroz et al., 2023) but with a particular emphasis on the recent research line in the Wasserstein geometric viewpoint of this family (Lambert et al., 2022; Diao et al., 2023). Regarding the target $\pi$, we assume access to second order gradients, typically computed by automatic differentiation, similar to the above works. Gaussian VI offers strong statistical guarantees at the optimal solution (Katsevich & Rigollet, 2024), offers an easy way of modelling dependencies between the variables and, thanks to the Bernstein-von Mises theorem (Van der Vaart, 2000), becomes asymptotically exact for Bayesian inference at the limit of infinite observations.

Recently, there has been emerging interest in Gaussian VI with a new geometric Riemannian optimization perspective (Lambert et al., 2022; Diao et al., 2023). The family of non-degenerate Gaus-

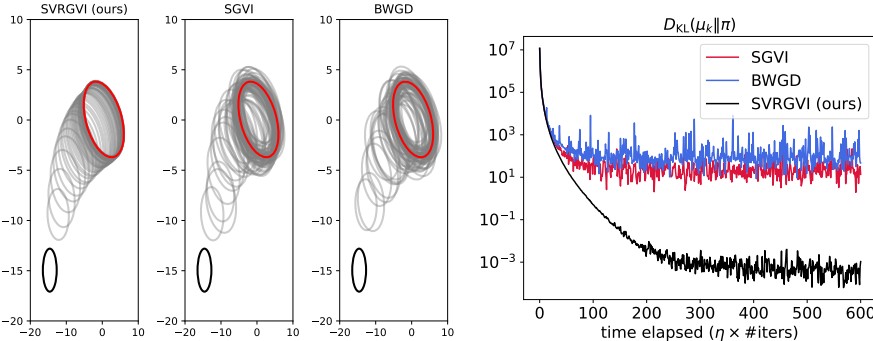

Figure 1: **Left**: Optimization trajectories of our method compared to SGVI (Diao et al., 2023) and BWGD (Lambert et al., 2022). The target is a 50-dimensional Gaussian distribution, visualized here via the marginal distributions of the first two coordinates. Each ellipse represents a contour of a Gaussian: the black is the initial distribution, the red is the target, and the greys are intermediate steps. Our method is dramatically more stable and finds a more accurate final approximation. **Right**: the corresponding KL divergence, confirming our method is orders of magnitude more accurate.

sian distributions can be parameterized by its mean and covariance matrix, $\mu_\theta$ with $\theta = (m, \Sigma)$, henceforth denoted as $\Theta = \mathbb{R}^d \times \mathcal{S}_{++}^d$ where $\mathcal{S}_{++}^d$ is the set of $d \times d$ symmetric, positive definite matrices. Classical VI employs conventional optimization algorithms (Paisley et al., 2012; Titsias & Lázaro-Gredilla, 2014; Kucukelbir et al., 2017) to minimize the Kullback-Leibler (KL) divergence $D_{\mathrm{KL}}(\mu_\theta \| \pi)$ over the parameter space $\Theta$ equipped with the Euclidean geometry. Lambert et al. (2022) argue that because the optimization problem is over the space distributions, it is more natural to use the geometry of this space rather than the geometry of the parameter space. The space of Gaussian distributions has a rich, meaningful and tractable geometry known as Bures–Wasserstein (BW) geometry that benefits optimization. Lambert et al. (2022) subsequently established a theoretical framework for performing VI using the BW geometry, which we adopt in this paper.

Let $\pi(x) \propto \exp(-V(x))$ be the target distribution and consider the VI problem

$$\hat{\pi} \in \underset{\mu \in \mathrm{BW}(\mathbb{R}^d)}{\arg\min} \ D_{\mathrm{KL}}(\mu \| \pi), \tag{1}$$

where $\mathrm{BW}(\mathbb{R}^d)$ is the Bures–Wasserstein space of Gaussian distributions with non-degenerate covariance matrix. The BW space is a Riemannian manifold whose geodesic distance is the Bures–Wasserstein distance. This setting nicely interplays the theory of optimal transport, Wasserstein gradient flows, and variational inference. The optimization problem (1) can be reformulated as

$$\hat{\pi} \in \underset{\mu \in \mathrm{BW}(\mathbb{R}^d)}{\arg\min} \ \mathcal{F}(\mu), \quad \text{where} \quad \mathcal{F}(\mu) := \mathcal{E}_V(\mu) + \mathscr{H}(\mu). \tag{2}$$

Here, $\mathcal{E}_V(\mu) = \int V(x) d\mu(x)$ is the potential energy and $\mathscr{H}(\mu) = \int \log(\mu(x)) d\mu(x)$ is the negative entropy. A conceptual and established idea to minimize a functional $\mathcal{F}$ is to perform gradient flow on $\mathcal{F}$ with respect to the geometry of $\mathrm{BW}(\mathbb{R}^d)$. To be implementable, the flow must be discretized. Lambert et al. (2022) use forward Euler discretization, resulting in a scheme named Bures–Wasserstein stochastic gradient descent (BWGD).

Diao et al. (2023) remark that forward-backward (FB) Euler (Bauschke & Combettes, 2011) should be used instead due to the objective's composite nature and the entropy's non-smoothness. This method iteratively applies a forward step to the potential energy $\mathcal{E}_V$ and a backward step (proximal operator) to the negative entropy $\mathscr{H}$. They also observe that the backward step in the BW space has a closed-form solution (Wibisono, 2018). This is crucial because this step is known to be intractable (or computationally expensive) in the full Wasserstein space (Wibisono, 2018; Salim et al., 2020; Mokrov et al., 2021; Luu et al., 2024). Although the bottleneck of the FB Euler, which is the backward step, has been resolved in this case, the forward step becomes problematic where one has to compute the Bures–Wasserstein gradient of $\mathcal{E}_V$ instead of the "friendly" Wasserstein gradient that is just $\nabla V$. The Bures–Wasserstein gradient is not always available in closed form, i.e., at $\mu \in \mathrm{BW}(\mathbb{R}^d)$, it is given only implicitly by the map $x \mapsto \mathbb{E}_\mu \nabla V + (\mathbb{E}_\mu \nabla^2 V)(x - m_\mu)$ where $m_\mu$

is the mean of $\mu$ (Lambert et al., 2022). This is the orthogonal projection of the Wasserstein gradient onto a tangent space of the Bures–Wasserstein manifold (Chewi et al., 2024). For general $V$, these expectations are intractable even though the underlying distribution is a Gaussian. Diao et al. (2023) proposed using the Monte Carlo (MC) method with one sample to estimate these expectations at each iteration: sample $X \sim \mu$ and use $\nabla V(X)$ and $\nabla^2 V(X)$ as unbiased estimators for $\mathbb{E}_\mu \nabla V$ and $\mathbb{E}_\mu \nabla^2 V$, respectively. This scheme is called Stochastic Gaussian VI (SGVI).

The problem with SGVI building on this principle is that the Monte Carlo estimates needed for the BW gradient are typically too noisy, particularly in high dimensions, as shown in our experiments (Sect. 5). In practice, high-variance estimators require small step sizes, leading to slow and inefficient convergence. We resolve this fundamental limitation by proposing a variance-reduced estimator with minimal computational overhead while providing robust theoretical guarantees. Fig. 1 shows the improvement over SGVI and BWGD in practice. Bures–Wasserstein geometry offers a meaningful transition from the initial distribution to the target distribution, and our method follows the path smoothly and is particularly stable around the optimum.

**Contributions.** We propose a novel variance-reduced estimator for $\mathbb{E}_\mu \nabla V$ that does not use any extra samples, with minimal per-iteration computational overhead, using the control variates approach (Owen, 2013). Our idea is that the variational distribution $\mu$ should be similar to the target distribution $\pi(x) \propto \exp(-V(x))$ as $\mu$ gets closer and closer to $\pi$, so the density of $\mu$ can be used to construct a correlated control variate for the Monte-Carlo estimator $\nabla V(X)$. Sect. 3 presents the detailed construction and its rationale.

On the theoretical side, we derive the following insights:

Thm. 1    Under a mild smoothness assumption, we prove that there is a region around the optimal solution $\hat{\pi}$ where our estimator has guaranteed smaller variance than the MC estimator.

Thm. 2    If $V$ is strongly convex, we prove that the proposed estimator has a smaller variance than the MC estimator at every $\mu \in \mathrm{BW}(\mathbb{R}^d)$ whenever $\mu$ has sufficiently large (greater than a controllable threshold) variance.

We further show in Thm. 3 and Thm. 4 that whenever variance reduction happens along the algorithm's iterates, the effect will enter the convergence analysis and improve the optimization bounds derived in Diao et al. (2023). These theorems solidly back our proposed method.

On the practical side, we show that reusing the Cholesky decomposition of the covariance matrix (needed to sample from a multivariate Gaussian) keeps the computational overhead of the control variable negligible. Despite being only a minimal modification to the Monte Carlo estimator, the proposed estimator achieves significant improvements in our experiments.

## 2   Background

A function $f : \mathbb{R}^d \to \mathbb{R}$ is called L-smooth (or Lipschitz smooth) if $\|\nabla f(x) - \nabla f(y)\| \leq L\|x - y\|$ for all $x, y \in \mathbb{R}^d$. If $f$ is twice continuously differentiable, we define the Laplacian operator of $f$ as $\Delta f = \sum_{i=1}^d (\partial^2/\partial x_i^2)f$. For a random variable $\tau$, $\|\tau\|_\infty := \inf\{C : |\tau| \leq C \text{ almost surely}\}$.

### 2.1   Bures–Wasserstein geometry

We denote by $\mathcal{P}_2(\mathbb{R}^d)$ the space of probability measures $\mu$ over $\mathbb{R}^d$ with finite second-moment, i.e., $\int \|x\|^2 d\mu(x) < +\infty$. Equipped with the Wasserstein distance

$$W_2^2(\mu, \nu) = \inf_{\gamma \in \Gamma(\mu,\nu)} \int_{X \times X} \|x - y\|^2 d\gamma(x, y) \tag{3}$$

where $\Gamma(\mu, \nu)$ is the set of probability measures over $X \times X$ whose marginals are $\mu$ and $\nu$, the space $\mathcal{P}_2(\mathbb{R}^d)$ becomes the metric space called the Wasserstein space (Ambrosio et al., 2005). We call $\gamma \in \Gamma(\mu, \nu)$ a (transport) plan and any $\gamma$ that achieves the optimal value in (3) an optimal plan. A pair of random variables whose joint distribution is an optimal plan is called an optimal coupling (between $\mu$ and $\nu$). When $\mu$ is absolutely continuous with respect to the Lebesgue measure, Brenier theorem (Brenier, 1991) asserts that the optimal plan is unique and is given by $(I, T_\mu^\nu)_{\#}\mu$ where

$T^\nu_\mu = \nabla g$ for some convex function $g$. We call $T^\nu_\mu$ the optimal transport map from $\mu$ to $\nu$. Apart from being a metric space, the Wasserstein space also enjoys some nice properties of Riemannian geometry. Otto's calculus (Otto, 2001) endows the Wasserstein space with a formal Riemannian structure, facilitating gradient flows and optimization.

We denote by $\mathrm{BW}(\mathbb{R}^d)$ the space of Gaussian distributions with non-degenerate covariance matrices. The Wasserstein distance between two Gaussian distributions $p_0 = \mathcal{N}(m_0, \Sigma_0)$ and $p_1 = \mathcal{N}(m_1, \Sigma_1)$ is given in the closed-form formula $\mathcal{W}_2^2(p_0, p_1) = \|m_0 - m_1\|^2 + \mathcal{B}^2(\Sigma_0, \Sigma_1)$ where $\mathcal{B}^2(\Sigma_0, \Sigma_1) = \mathrm{Tr}(\Sigma_0 + \Sigma_1 - 2(\Sigma_0^{\frac{1}{2}} \Sigma_1 \Sigma_0^{\frac{1}{2}})^{\frac{1}{2}})$ is the Bures metric. The optimal transport map is also given in a closed form in this case: $T^{p_1}_{p_0}(x) = m_1 + \Sigma_0^{-\frac{1}{2}} \left( \Sigma_0^{\frac{1}{2}} \Sigma_1 \Sigma_0^{\frac{1}{2}} \right)^{\frac{1}{2}} \Sigma_0^{-\frac{1}{2}}(x - m_0)$.

The BW space is a geodesically convex subset of the Wasserstein space, meaning that a geodesic curve joining two Gaussians lies entirely inside the BW space. The BW space is a Riemannian manifold in its own right. Let $\mu = \mathcal{N}(m, \Sigma) \in \mathrm{BW}(\mathbb{R}^d)$, the tangent space of $\mathrm{BW}(\mathbb{R}^d)$ at $\mu$ is the space of symmetric affine maps denoted as $T_\mu \mathrm{BW}(\mathbb{R}^d) = \{x \mapsto S(x - m) + a \mid a \in \mathbb{R}^d, S \in \mathcal{S}^d\}$ where $\mathcal{S}^d$ is the space of symmetric $d \times d$ matrices. The Riemannian metric defined using the inner product of elements in this tangent space is identified as the $L^2(\mu)$ inner product restricted to this space. Given $U, V \in T_\mu \mathrm{BW}(\mathbb{R}^d)$, the metric is $\langle U, V \rangle_\mu := \int \langle U(x), V(x) \rangle d\mu(x)$. This Riemannian metric induces the geodesic distance in $\mathrm{BW}(\mathbb{R}^d)$ that is given by the Wasserstein distance. We refer to Altschuler et al. (2021) for further discussions on BW geometry.

## 2.2 STOCHASTIC GAUSSIAN VI

We refer to Diao et al. (2023) for a detailed discussion and relevant terminologies. We briefly explain stochastic Gaussian VI to motivate our proposed variance reduction. Recall from (2) that we aim to minimize $\mathcal{F}(\mu) = \mathscr{H}(\mu) + \mathcal{E}_V(\mu)$ over $\mathrm{BW}(\mathbb{R}^d)$. At the optimum of $\mathcal{F}$, $\hat{\pi} = \mathcal{N}(\hat{m}, \hat{\Sigma})$, first-order optimality reads (Opper & Archambeau, 2009; Lambert et al., 2022; Diao et al., 2023)

$$\mathbb{E}_{\hat{\pi}} \nabla V = 0 \quad \text{and} \quad \mathbb{E}_{\hat{\pi}} \nabla^2 V = \hat{\Sigma}^{-1} \tag{4}$$

which is derived by zeroing the Bures–Wasserstein gradient of the objective function.

A natural idea to minimize $\mathcal{F}$ over $\mathrm{BW}(\mathbb{R}^d)$ is to perform gradient flow on $\mathcal{F}$ using the BW geometry of $\mathrm{BW}(\mathbb{R}^d)$. When the gradient flow is applied over the entire Wasserstein space $\mathcal{P}_2(\mathbb{R}^d)$, it corresponds to the Langevin diffusion (Jordan et al., 1998), with one of its discretizations being an MCMC method called the unadjusted Langevin algorithm (Roberts & Tweedie, 1996). When restricted to $\mathrm{BW}(\mathbb{R}^d)$, the gradient flow can be formulated using Riemannian geometry (Do Carmo, 1992), as $\mathrm{BW}(\mathbb{R}^d)$ forms a true Riemannian manifold. This flow is a curve of Gaussian distributions, characterized by the time-dependent evolution of their mean and covariance matrix. Recently, Lambert et al. (2022) showed that this evolution is governed by Särkkä's ODEs developed in the context of variational Kalman filtering (Särkkä, 2007).

The negative entropy $\mathscr{H}$ is convex along generalized geodesics but it is a nonsmooth functional. If $V$ is smooth, it induces the smoothness of $\mathcal{E}_V$. Therefore, it is natural to apply forward-backward Euler that alternates between two steps: at iteration $k$,

$$\mu_{k+\frac{1}{2}} = (I - \eta \nabla_{\mathrm{BW}} \mathcal{E}_V(\mu_k))_\# \mu_k \quad \triangleleft \text{forward step}$$

$$\mu_{k+1} = \underset{\mu \in \mathrm{BW}(\mathbb{R}^d)}{\arg\min} \left\{ \mathscr{H}(\mu) + \frac{1}{2\eta} W_2^2 \left( \mu, \mu_{k+\frac{1}{2}} \right) \right\} \quad \triangleleft \text{backward step}$$

where $\nabla_{\mathrm{BW}}$ denotes the Bures–Wasserstein gradient. The backward step is also known as the proximal step in the optimization literature or the JKO (Jordan, Kinderlehrer, and Otto) step (with restriction in $\mathrm{BW}(\mathbb{R}^d)$) in the context of Wasserstein gradient flow (Jordan et al., 1998). The backward step is intractable in the full Wasserstein space and hence requires (oftentimes expensive) numerical approximations (Mokrov et al., 2021; Luu et al., 2024). On the other hand, if restricted to $\mathrm{BW}(\mathbb{R}^d)$, this step admits a closed-form solution (Wibisono, 2018): let $\mu_{k+\frac{1}{2}} = \mathcal{N}(m_{k+\frac{1}{2}}, \Sigma_{k+\frac{1}{2}})$, then $\mu_{k+1}$ is a Gaussian distribution with mean $m_{k+1} = m_{k+\frac{1}{2}}$ and variance matrix $\Sigma_{k+1} = \frac{1}{2} \left( \Sigma_{k+\frac{1}{2}} + 2\eta I + [\Sigma_{k+\frac{1}{2}}(\Sigma_{k+\frac{1}{2}} + 4\eta I)]^{\frac{1}{2}} \right)$. This tractability of the backward is the main motivation for Diao et al. (2023) to study and develop FB Euler in this scenario. The forward step, however,

is not always analytically available since the BW gradient of $\mathcal{E}_V$, at iterate $k$,

$$\nabla_{\mathrm{BW}}\mathcal{E}_V(\mu_k) : x \mapsto \mathbb{E}_{\mu_k}\nabla V + (\mathbb{E}_{\mu_k}\nabla^2 V)(x - m_k),$$

involves intractable expectations. Diao et al. (2023) propose using Monte Carlo approximation for these expectations: sample $X_k \sim \mu_k$ and use $b_k := \nabla V(X_k)$ and $S_k := \nabla^2 V(X_k)$ as unbiased estimators for $\mathbb{E}_{\mu_k}\nabla V$ and $\mathbb{E}_{\mu_k}\nabla^2 V$, respectively.

## 3 STOCHASTIC VARIANCE-REDUCED GAUSSIAN VI

We present our ideas on constructing stochastic variance-reduced estimators from first principles. We recall from Sect. 2.2 that stochastic Gaussian VI approximates, at iteration $k$,

$$\mathbb{E}_{\mu_k}\nabla V \approx b_k := \nabla V(X_k) \quad \text{and} \quad \mathbb{E}_{\mu_k}\nabla^2 V \approx S_k := \nabla^2 V(X_k) \quad \text{where } X_k \sim \mu_k. \quad (5)$$

These estimators are typically noisy. Any number of MC samples can be used, but already one is unbiased and proposed by earlier works; we also focus on the single-sample case for computational efficiency. We aim to design better unbiased estimators for either $\mathbb{E}_{\mu_k}\nabla V$ or $\mathbb{E}_{\mu_k}\nabla^2 V$ in the sense that their variances are smaller than those of $b_k$ and $S_k$, building on the control variates approach (Owen, 2013); Also see the discussions in Defazio et al. (2014); Luu (2022).

Let us first describe briefly the core idea of control variates in helping reduce the variance. Let $\theta$ be the quantity of interest and $X$ be an unbiased estimator for $\theta$, i.e., $\mathbb{E}X = \theta$. A *control variate* is a random variable $Y$ with a known mean so that $Y$ is correlated with $X$. The random variable $Z = X + c(\mathbb{E}Y - Y)$, where $c \in \mathbb{R}$, is then an *unbiased estimator* for $\theta$. The variance of $Z$ is

$$\mathrm{Var}Z = \mathrm{Var}X + c^2\mathrm{Var}Y - 2c\mathrm{Cov}(X,Y). \quad (6)$$

If $X,Y$ are highly correlated in the sense that $2\mathrm{Cov}(X,Y) > \mathrm{Var}Y$, we immediately get $\mathrm{Var}Z < \mathrm{Var}X$ for any $c \in (0,1]$. So, we achieve a reduction in variance by using $Z$. On the other hand, if $X,Y$ are correlated ($\mathrm{Cov}(X,Y) > 0$) but not highly correlated, we can also obtain variance reduction effects whenever $c$ is positive and small enough. Furthermore, given the parabolic form with respect to $c$ in (6), one can pinpoint the optimal value of $c$ is $c^* := \mathrm{Cov}(X,Y)/\mathrm{Var}(Y)$, resulting in the maximal variance reduction $\mathrm{Var}Z = (1 - \mathrm{Corr}(X,Y)^2)\mathrm{Var}X < \mathrm{Var}X$ where $\mathrm{Corr}(X,Y)$ denotes correlation between $X$ and $Y$.

We now return to our problem and seek variance-reduced estimators of the forms

$$\tilde{b}_k := \nabla V(X_k) + c(\mathbb{E}(Z_k) - Z_k) \quad \text{and} \quad \tilde{S}_k := \nabla^2 V(X_k) + d(\mathbb{E}(W_k) - W_k)$$

where $c, d > 0$ and $Z_k, W_k$ are a random vector and a random matrix, respectively. Let us first focus on $\tilde{b}_k$. As discussed, $Z_k$ should be (element-wise) highly correlated with $\nabla V(X_k)$ while $\mathbb{E}(Z_k)$ remains efficiently computable. We look for $Z_k = \nabla U(X_k)$ so that $\nabla U$ is as close to $\nabla V$ as possible. We are in the context of approximating $\pi(x)$ by the VI distribution $\mu_k = \mathcal{N}(m_k, \Sigma_k)$, so it is natural to expect that

$$-\nabla V(x) = \nabla \log \pi(x) \approx \nabla \log f(x; m_k, \Sigma_k) = -\Sigma_k^{-1}(x - m_k).$$

where $f(x; m_k, \Sigma_k) \propto \exp\left(-\frac{1}{2}(x - m_k)^\top \Sigma_k^{-1}(x - m_k)\right)$ is the PDF of $\mu_k$. Therefore, we propose using $Z_k = \Sigma_k^{-1}(X_k - m_k)$ as a control variate. We have $\mathbb{E}(Z_k) = 0$ since $\mathbb{E}(X_k) = m_k$. It is worth noting that $Z_k$ is known as the Stein/Hyvärinen score (Hyvärinen, 2005) of $\mu_k$. The estimator $\tilde{b}_k$ then becomes $\tilde{b}_k := \nabla V(X_k) - c\Sigma_k^{-1}(X_k - m_k)$. By applying the same reasoning to $\tilde{S}_k$, we can immediately conclude that $W_k$ is deterministic and equals $\Sigma_k^{-1}$. Consequently, the control variate does not affect $S_k$; we keep the standard estimator. We derive Stochastic variance-reduced Gaussian VI (SVRGVI) as in Alg. 1 (we will discuss more about the choice of $c_k$ in Sect. 4). Note that the only difference between Alg. 1 and the SGVI in Diao et al. (2023) is the estimator $\tilde{b}_k$, where the difference is highlighted in **blue**.

Fig. 2 (left) demonstrates that our proposed estimator (with $c = 0.9$) achieves lower variance compared to the standard MC estimator, while both remain unbiased estimators of $\mathbb{E}_\mu\nabla V$. In Fig. 2 (right), we vary $c$ from $0$ to $2$ and calculate the empirical variance of our estimator, revealing a parabolic pattern. Note that when $c = 0$, the estimator reduces to the standard estiator, and for all values of $c \in (0,2)$, our proposed estimator consistently exhibits lower variance, with an optimal value of $c$ around $1$. At this optimal $c$, the variance is reduced roughly by a factor of $10$. We provide theoretical justification for these empirical observations in Sect. 4.

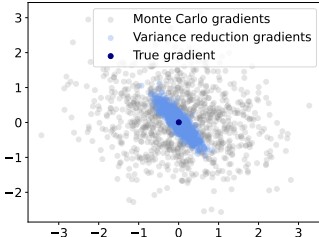 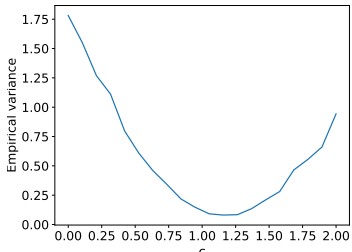

Figure 2: **Left**: $\pi$ is a Gaussian, VI distribution $\mu$ is in the neighborhood of $\pi$. In this case, the true gradient, i.e., the expectation $\mathbb{E}_\mu \nabla V$, can be computed exactly (in **navy blue**). Our proposed estimator with $c = 0.9$ (**light blue**) has a smaller variance than the Monte Carlo estimator (**grey**). These are 1,000 samples for each estimator, generated by drawing from $\mu$ and substituting the values into the respective estimator formulas. **Right**: The empirical variance of our proposed estimator when $c$ varies from 0 to 2. Note that $c = 0$ corresponds to the Monte Carlo estimator.

**Minimal extra computational cost** Despite involving calculating the inverse of the covariance matrix, the computational overhead is small. Sampling from multivariate normal in step 1 in Alg. 1 typically requires obtaining the Cholesky factor of the covariance matrix, which is $O(d^3)$ (Rasmussen & Williams, 2006) . With the Cholesky factor, obtaining the solution of the inverse of the matrix times a vector is $O(d^2)$ (Rasmussen & Williams, 2006). As such, we can reuse this obtained Cholesky factor in step 1 to compute the inverse in step 2, which implies that the estimator adds an overhead of $O(d^2)$, which is naturally dominated by the $O(d^3)$ complexity of the original algorithm.

---

**Algorithm 1** Stochastic variance-reduced Gaussian Variational Inference (SVRGVI)

---

**Input:** Target distribution $\pi(x) \propto \exp(-V(x))$, initial distribution $\mu_0 = \mathcal{N}(m_0, \Sigma_0)$, step size $\eta > 0$, number of steps $N$, sequence of control variate parameters $\{c_k\}_{k=0}^{N-1}$ where $c_k \in (0, 1], \forall k \in \{0, 1, \ldots, N-1\}$
**for** $k = 0$ to $N - 1$ **do**
  1. Draw one sample $X_k \sim \mathcal{N}(m_k, \Sigma_k)$
  2. Compute estimators: $\tilde{b}_k \leftarrow \nabla V(X_k) - c_k \Sigma_k^{-1}(X_k - m_k)$ and $S_k \leftarrow \nabla^2 V(X_k)$
  3. Update mean and covariance matrix:
    $m_{k+1} \leftarrow m_k - \eta \tilde{b}_k$
    $M_{k+1} \leftarrow I - \eta S_k$
    $\Sigma_{k+\frac{1}{2}} \leftarrow M_{k+1} \Sigma_k M_{k+1}$
    $\Sigma_{k+1} = \frac{1}{2} \left( \Sigma_{k+\frac{1}{2}} + 2\eta I + \left[ \Sigma_{k+\frac{1}{2}} (\Sigma_{k+\frac{1}{2}} + 4\eta I) \right]^{\frac{1}{2}} \right)$
**end for**
**Output:** $\mu_N = \mathcal{N}(m_N, \Sigma_N)$

---

## 4 THEORY

In Sect. 3, we argued that, in the context of variational inference, as $\mu_k$ iteratively gets closer $\pi$, $\nabla V(X_k)$ is then (highly) correlated to $\Sigma_k^{-1}(X_k - m_k)$, and hence we obtain a variance reduction effect. This argument leads to the construction of the control variate in Alg. 1. One might question whether this approach remains effective when the target distribution $\pi$ is significantly distant from the BW space. Because we are constrained to the BW space, the best we can do is to get closer to $\hat{\pi}$ which is the optimal solution to the problem (1). However, $\hat{\pi}$ might still look very different from $\pi$. Notably, in Thm. 1, we rigorously show that within a certain neighbourhood of $\hat{\pi}$ (to be defined later), our proposed estimator consistently reduces variance, regardless of how different $\pi$ is to a Gaussian distribution. Let us first introduce Lem. 1 to pave the way for Thm. 1 and also to discuss

the optimal $c$ in the control variate. In Lem. 1, we compute the variance of the proposed estimator by leveraging multidimensional Stein's lemma (Lin et al., 2019).

**Lemma 1** *Assume that $V$ is continuously differentiable. Let $\mu = \mathcal{N}(m, \Sigma) \in \mathrm{BW}(\mathbb{R}^d)$. Then,*

$$\underbrace{\mathbb{E}\|\nabla V(X) - c\Sigma^{-1}(X-m) - \mathbb{E}\nabla V(X)\|^2}_{\text{variance of our estimator}}$$

$$= \underbrace{\mathbb{E}\|\nabla V(X) - \mathbb{E}\nabla V(X)\|^2}_{\text{variance of the Monte-Carlo estimator}} + \underbrace{c^2\operatorname{Tr}(\Sigma^{-1}) - 2c\operatorname{Tr}(\mathbb{E}\nabla^2 V(X))}_{\text{extra term}}, \text{ where } X \sim \mu.$$

Proof of Lem. 1 is given in Appendix A.1. Lem. 1 compares the variance of the proposed estimator and the Monte Carlo estimator at a given $\mu \in \mathrm{BW}(\mathbb{R}^d)$. Recall that the first-order optimality condition (4) of $\hat{\pi}$ reads $\mathbb{E}_{\hat{\pi}}\nabla^2 V = \hat{\Sigma}^{-1}$. Consequently, at $\hat{\pi}$, the *extra term* in Lem. 1 is simplified as $c(c-2)\operatorname{Tr}(\hat{\Sigma}^{-1})$ which is negative whenever $c \in (0, 2)$ and minimized for $c = 1$. Therefore, at $\hat{\pi}$, our estimator is always better than the Monte Carlo estimator for $c \in (0, 2)$.

**Remark 1** *A practical merit of Lem. 1 is that it implies the optimal value for $c$ to get maximum variance reduction at $\mu$ is $c^* = \operatorname{Tr}(\mathbb{E}_\mu\nabla^2 V)/\operatorname{Tr}(\Sigma^{-1})$. Applying this to Alg. 1, we can pick the adaptive sequence $\{c_k\}$ as $c_k^* = \frac{\operatorname{Tr}(\mathbb{E}_{\mu_k}\nabla^2 V)}{\operatorname{Tr}(\Sigma_k^{-1})} \approx \frac{\operatorname{Tr}(S_k)}{\operatorname{Tr}(\Sigma_k^{-1})} := c_k$. Again, this computation of $c_k$ incurs a negligible extra cost to Alg. 1. We also remark that around $\hat{\pi}$, optimality condition (4) implies the optimal value $c^*$ indeed is around $1$.*

In Thm. 1, we further show that when the Laplacian $\Delta V$ is smooth, the proposed estimator has a smaller variance than the Monte Carlo estimator in a region around $\hat{\pi}$.

**Theorem 1 (Variance reduction around the optimal solution)** *Assume that the Laplacian $\Delta V$ is $\ell$-smooth. For any control variate coefficient $c \in (0, 2)$, define the region around $\hat{\pi} = \mathcal{N}(\hat{m}, \hat{\Sigma})$:*

$$\mathcal{V}(\hat{\pi}, r) = \{\mu = \mathcal{N}(m, \Sigma) : 2\ell W_2(\mu, \hat{\pi}) + c|\operatorname{Tr}(\Sigma^{-1}) - \operatorname{Tr}(\hat{\Sigma}^{-1})| < r\}$$

*where $r = (2-c)\operatorname{Tr}(\hat{\Sigma}^{-1}) > 0$ is the region's radius. For any $\mu \in \mathcal{V}(\hat{\pi}, r)$, the proposed estimator has a smaller variance than the Monte Carlo estimator.*

Proof of Thm. 1 is given in Appendix A.2 with the main idea being that the smoothness of the Laplacian $\Delta V$ propagates the improvement of the proposed estimator at $\hat{\pi}$ to its neighbourhood. We additionally observe that, for small $c > 0$, the region $\mathcal{V}(\hat{\pi}, r)$ effectively reduces to the Wasserstein ball $\mathcal{B}(\hat{\pi}, \ell^{-1}\operatorname{Tr}(\hat{\Sigma}^{-1}))$.

Thm. 1 applies to arbitrary $\pi$, only requiring a mild smoothness condition of its second derivative. In the next theorem, we show that when $\pi$ is strongly log-concave ($\pi$ is now more similar to a Gaussian), variance reduction happens not only around $\hat{\pi}$ but also in many regions of interest.

**Theorem 2 (Variance reduction at large-variance distributions)** *If $V$ is $\alpha$-strongly convex for some $\alpha > 0$, for any control variate $c > 0$, the proposed estimator has a smaller variance than the Monte Carlo estimator at every $\mu = \mathcal{N}(m, \Sigma)$ whenever $\operatorname{Tr}(\Sigma^{-1}) < \frac{2\alpha d}{c}$.*

Proof of Thm. 2 is given in Appendix A.3.

**Remark 2** *A consequence of Thm. 2 is that for strongly convex $V$ we can obtain variance reduction at each iteration in Alg. 1 by setting $c_k \in (0, 2\alpha d/\operatorname{Tr}(\Sigma_k^{-1}))$. We can extend this result to the case of convex but non-strongly-convex $V$, i.e., $\alpha = 0$. We first show that: for any Gaussian $\mathcal{N}(m, \Sigma)$, $\operatorname{Tr}(\mathbb{E}_{\mathcal{N}(m,\Sigma)}\nabla^2 V) > 0$. We can then obtain variance reduction at each iteration by setting $c_k \in (0, 2\operatorname{Tr}(\mathbb{E}_{\mathcal{N}(m_k,\Sigma_k)}\nabla^2 V/\operatorname{Tr}(\Sigma_k^{-1}))$. Note that this upper bound of $c_k$ is less explicit as the strongly convex case. However, the point is that variance reduction still happens in this case given that $c_k$ is small enough. See Appendix A.3 for the proof.*

We further show in Thm. 3 and Thm. 4 that whenever variance reduction happens along the algorithm's iterates, the effect will propagate to the convergence analyses of Diao et al. (2023) and

improve their theoretical bounds. Therefore, combining with Thm. 1 and Thm. 2, the overall theory strongly favours SVRGVI over SGVI.

Let $\mathcal{P}_k$ denote the information up to the beginning of iteration $k$, i.e., it is the $\sigma$-algebra given by $\mathcal{P}_k = \sigma(X_0, X_1, \ldots, X_{k-1})$ for $k \in \{1, 2, \ldots, N-1\}$ and $\mathcal{P}_0$ is, by convention, the trivial $\sigma$-algebra with no information. Assuming variance reduction occurs along the algorithm's iterates, i.e., for $k = 0, 1, \ldots, N-1$, it holds

$$\mathbb{E}\left(\|\nabla V(X_k) - c_k \Sigma_k^{-1}(X_k - m_k) - \mathbb{E}_{\mu_k} \nabla V\|^2 | \mathcal{P}_k\right) \leq \tau_k \mathbb{E}\left(\|\nabla V(X_k) - \mathbb{E}_{\mu_k} \nabla V\|^2 | \mathcal{P}_k\right) \quad (7)$$

where $\tau_k < 1$ almost surely. We recall from Remark 2 that for convex $V$, (7) holds such that $\tau_k < 1$ almost surely for all $k$ if $c_k$ is well-chosen, and it follows that $\mathbb{E}\tau_k < 1$ and $\|\tau_k\|_\infty \leq 1$ for all $k$. We can further guarantee $\|\tau_k\|_\infty < 1$ for all $k$ under mild boundedness assumptions (Appendix A.4).

Under condition 7, we now show the improved bounds. Similar to Diao et al. (2023), we consider log-concave and strongly-log-concave sampling, meaning that $V$ is assumed to be convex and strongly convex, respectively.

**Theorem 3 (Convex case)** *Suppose that $V$ is convex and $\beta$-smooth and the step size $0 < \eta \leq \frac{1}{2\beta}$. If variance reduction happens for $k = 0, 1, \ldots, N-1$, then,*

$$\mathbb{E} \min_{k=1,2,\ldots,N} \mathcal{F}(\mu_k) - \mathcal{F}(\hat{\pi}) \lesssim \frac{e}{1 + \frac{C\eta^2(1-\tau_{\max,\infty})}{2}} \left(\frac{1}{2\eta N} + \frac{C\eta}{2}\right) W_2^2(\mu_0, \hat{\pi}) + 3\eta\beta d(1 + \tau_{\max,E})$$

*where $\tau_{\max,\infty} := \max_{i=\overline{0,N-1}} \|\tau_i\|_\infty$, $\tau_{\max,E} = \max_{i=\overline{0,N-1}} \mathbb{E}\tau_i$, $e \approx 2.718$ is the Euler's number, $C = 24\beta^3 \lambda_{\max}(\hat{\Sigma})$, and $\lesssim$ is asymptotically at the limit of small $\eta$.*

Proof of Thm. 3 is given in Appendix A.5.

**Theorem 4 (Strongly convex case)** *Suppose that $V$ is $\alpha$-strongly-convex with $\alpha > 0$, and $0 < \eta \leq \frac{\alpha^2}{48\beta^3}$. If variance reduction happens for $k = 0, 1, \ldots, N-1$, then*

$$\mathbb{E}W_2^2(\mu_N, \hat{\pi}) \lesssim \exp\left(-\frac{N(3 - \tau_{\max,\infty})\eta\alpha}{4}\right) W_2^2(\mu_0, \hat{\pi}) + \frac{24(1 + \tau_{\max,E})\beta\eta d}{(3 - \tau_{\max,\infty})\alpha} \quad (8)$$

*where $\tau_{\max,\infty}$ and $\tau_{\max,E}$ are defined in Theorem 3, and $\lesssim$ is asymptotically at the limit of small $\eta$.*

Proof of Thm. 4 is given in Appendix A.6.

**Remark 3** *We recall the corresponding bounds for SGVI in Diao et al. (2023, Thm 5.7, Thm. 5.8)*

- ***Convex.*** $\mathbb{E}\left(\min_{k=1,2,\ldots,N} \mathcal{F}(\mu_k)\right) - \mathcal{F}(\hat{\pi}) \lesssim \frac{eW_2^2(\mu_0,\hat{\pi})}{2N\eta} + \frac{eC\eta}{2} W_2^2(\mu_0, \hat{\pi}) + 6\beta\eta d.$[1]

- ***Strongly convex.*** $\mathbb{E}W_2^2(\mu_N, \hat{\pi}) \lesssim \exp\left(-\frac{\alpha N\eta}{2}\right) W_2^2(\mu_0, \hat{\pi}) + \frac{24\beta\eta d}{\alpha}.$

*Putting side-by-side, we see that Thm. 3 and Thm. 4 improve all coefficients of these bounds. In particular, the scale-down involving $d$ is expected to help in high dimensions. It is also worth noting that even when we set $\tau_{\max,\infty} = 0$, the noise terms in the bounds of Thm. 3 and Thm. 4 would not disappear because of another source of randomness coming from $S_k$.*

## 5 EXPERIMENTS

We demonstrate the method in a collection of controlled problems, comparing it against the recent methods for VI in the BW manifold, namely BWGD (Lambert et al., 2022) and SGVI (Diao et al., 2023) [2]. We set the step size to 1 for all algorithms (for an experiment on varying step size, see

---

[1]With a minor correction to the coefficients in SGVI's bound.
[2]Our code is available at https://github.com/MCS-hub/vr25

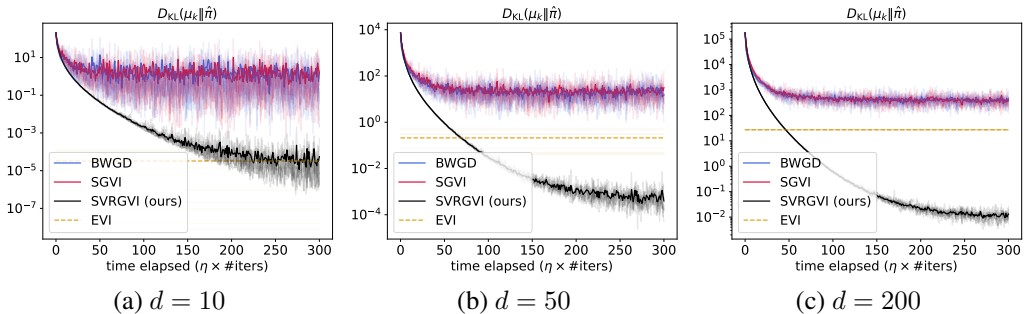

Figure 3: KL divergence for Gaussian targets of varying dimensionality.

Appendix B.7), fix the covariate coefficient $c = 0.9$ (for an experiment on the effect of $c$, see Appendix B.6), and show results for 10 runs, with bold line showing the average performance. The comparisons are shown as convergence curves, as the per-iteration cost of all methods is almost identical. We also compare against a full-rank Gaussian approximation optimised in the Euclidean geometry (denoted as EVI), using low-variance reparameterization gradients of Roeder et al. (2017) with ADAM optimizer, and Laplace approximation that does not optimize the KL divergence but fits a Gaussian distribution at the target mode; see Appendix B for details. As the per-iteration cost of these methods is different from the BW methods, we only report the final accuracy for carefully optimized approximations to show how the BW methods compare against commonly used algorithms. The Laplace approximation is omitted for Gaussian targets as it is optimal by definition.

**Gaussian targets** We randomly generate the means and covariances for a multivariate Gaussian target distribution $\pi$, considering dimensions of $\{10, 50, 200\}$. Fig. 3 demonstrates consistent significant improvement over SGVI and BWGD. For example, for $d = 200$, the the difference between SVRGVI and SGVI/BWGD is 5 orders of magnitude, $10^{-2}$ versus $10^3$. Fig. 1 shows visually the marginals for $d = 50$, providing an interpretation of the improvement seen in KL-divergence. We also clearly outperform EVI in higher dimensions, unlike previous BW methods.

**Student's t targets** We consider a multivariate Student's t target with a degree of freedom of 4 in 200 dimensions. Fig. 4 (a) shows that our algorithm is again clearly the best. BWGD is not stable and, on average, performs worse than even the Laplace approximation.

**Bayesian logistic regression** We consider a Bayesian logistic regression with a flat prior as in Diao et al. (2023): given a set of covariates $X_i \sim \mathcal{N}(0, I_d)$ for $i = 1, 2, \ldots, n$, consider

$$Y_i | X_i, \theta \sim \text{Bernoulli}(\sigma(\langle \theta, X_i \rangle)), \text{ where } \sigma \text{ is the sigmoid function.}$$

The negative log posterior is $V(\theta) = \sum_{i=1}^{n} \left[ \ln(1 + e^{\langle \theta, X_i \rangle}) - Y_i \langle \theta, X_i \rangle \right]$. The model consists of $n = 1000$ data points $(X_i, Y_i)$ with dimension $d = 200$. The optimal solution is unknown in this case, so we cannot plot the KL divergence along the iterations. Instead, we estimate the objective function of the problem (2), $\mathcal{F}(\mu_k)$, by drawing samples from $\mu_k$. We denote by $\mu_{\text{best}}$ the distribution that obtains the smallest $\mathcal{F}$ among all iterations of all algorithms, comparing against that. Fig 4 (b) shows the proposed method is again the most accurate.

We also measured the variance along iterations in Appendix B.4, showing that we achieved variance reduction empirically. We showed in Appendix B.5 that the proposed method is more efficient than minibatching (whether using i.i.d. or quasi Monte-Carlo samples).

## 6 DISCUSSION

Various variance reduction techniques have been broadly studied in the VI literature, but mainly for methods operating in the Euclidean parameter space. Our work resembles in nature the seminal work of Roeder et al. (2017) that demonstrated how the variance of gradient estimators for VI can be dramatically reduced by a single-line change in the algorithm: We also propose a minor

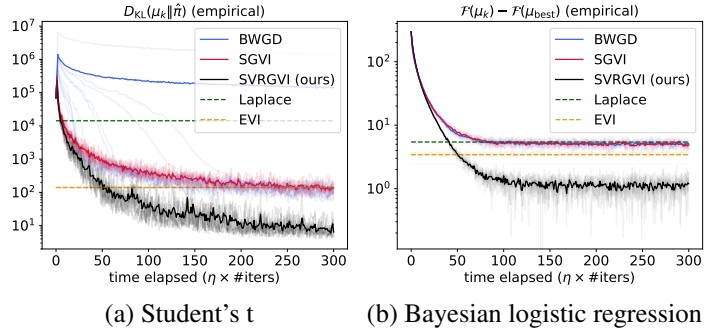

(a) Student's t       (b) Bayesian logistic regression

Figure 4: Performance of algorithms for Student's t target and Bayesian logistic regression.

modification that dramatically improves the accuracy, and should always be used. A high-level similarity lies in the heuristic that the VI distribution resembles the target distribution, allowing it to be used to construct control variates. Perhaps surprisingly, Yi & Liu (2023) recently showed that the continuous-time flow of Roeder et al. (2017) for Gaussian VI in the Euclidean geometry (by reparameterizing the covariance matrix as $\Sigma = SS^{\top}$) also follows Särkkä's ODEs. In other words, those BW methods and Roeder et al. (2017) are different discretizations of the same dynamics. Recent works (Kim et al., 2023; Domke et al., 2024) have advanced our understanding of variance reduction in Euclidean VI, demonstrating strong convergence and extending beyond traditional control variate approaches. Adopting these new techniques in the BW setting is a promising research direction.

In the BW space, Diao (2023) considers variance reduction for large-sum structures based on the nested-loop idea by Johnson & Zhang (2013) to reduce the stochasticity of the minibatch sampling. In contrast, our method addresses the stochasticity arising from Gaussian sampling from the VI.

Even though our experiments focused on synthetic targets, they expanded on the previous experimentation of VI optimized in the BW space. We confirm the finding of Diao et al. (2023) that BWGD and SGVI are effectively identical except for the instability of the former, but now show how their performance degrades in higher dimensions while our algorithm remains effective.

With the exception of the vastly improved accuracy due to the significantly lower variance of the gradient estimators, our method retains all qualitative characteristics of the previous BW methods, both positive and negative. That is, we retain the theoretical convergence guarantees and asymptotic optimality for posterior inference, but also the cubic computational cost due to requiring the Hessian of the log-target and the limitation to Gaussian approximations by construction. As highlighted by Xu & Campbell (2022) and Quiroz et al. (2023), there are tasks for which Gaussian approximations are highly relevant due to efficiently capturing the correlations.

## 7   CONCLUSION

Our main result is showing that the methods learning a variational approximation by direct optimization of the approximating distribution in the Bures–Wasserstein space of Gaussians can be made practical. The previous works by Lambert et al. (2022) and Diao et al. (2023) introduced the key idea and the algorithms with strong theoretical guarantees. However, they are prone to high variance from Monte Carlo approximations, limiting the impact. Our variance reduction technique that requires only a minor modification for the SVGI algorithm completely resolves this issue, resulting in extremely stable learning.

We demonstrated substantial variance reduction and showed that this reduction results in orders of magnitude improvement in final approximation accuracy, over both the previous BW methods and examples of parameter-space algorithms. This improvement comes with provable variance reduction in the neighborhood of the optimal solution and for all distributions with sufficiently large variance in the case of strong log-concave targets, and hence the proposed variance reduction technique should always be used.

ACKNOWLEDGMENTS

This work is supported by the Research Council of Finland's Flagship programme: Finnish Center for Artificial Intelligence (FCAI), and additionally by grants 363317, 345811 and 348952. We thank Kai Puolamäki for the discussions. The authors wish to acknowledge CSC – IT Center for Science, Finland, for computational resources. We thank the anonymous reviewers for their insightful comments and suggestions.

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

## A  THEORY

### A.1  PROOF OF LEMMA 1

For each $\mu = \mathcal{N}(m, \Sigma) \in \mathrm{BW}(\mathbb{R}^d)$ and $c > 0$, we denote

$$\mathcal{Q}(\mu) = \mathbb{E}\|\nabla V(X) - \mathbb{E}\nabla V(X)\|^2 - \mathbb{E}\|\nabla V(X) - c\Sigma^{-1}(X - m) - \mathbb{E}\nabla V(X)\|^2, \quad X \sim \mu,$$

which is the difference between the variances of the Monte Carlo estimator and our proposed esti-mator. We want $\mathcal{Q}(\mu) > 0$. Simple algebras simplify $\mathcal{Q}$ as

$$\mathcal{Q}(\mu) = 2c\mathbb{E}\langle\nabla V(X) - \mathbb{E}\nabla V(X), \Sigma^{-1}(X - m)\rangle - c^2\mathbb{E}\|\Sigma^{-1}(X - m)\|^2.$$

Recall a standard result: if $X \sim \mathcal{N}(m, \Sigma)$, then its affine transformation $W = AX + b$ has the distribution $\mathcal{N}(Am+b, A\Sigma A^\top)$. Applying this result, $W := \Sigma^{-1}(X-m) \sim \mathcal{N}(0, \Sigma^{-1})$. Therefore

$$\mathbb{E}\|W\|^2 = \sum_{i=1}^{d} \mathbb{E}W_i^2 = \mathrm{Tr}(\Sigma^{-1}).$$

On the other hand,

$$\begin{aligned}
&\mathbb{E}\langle\nabla V(X) - \mathbb{E}\nabla V(X), \Sigma^{-1}(X - m)\rangle \\
&= \mathbb{E}\langle\nabla V(X), \Sigma^{-1}(X - m)\rangle - \mathbb{E}\langle\mathbb{E}\nabla V(X), \Sigma^{-1}(X - m)\rangle \\
&= \mathbb{E}\langle\nabla V(X), \Sigma^{-1}(X - m)\rangle - \langle\mathbb{E}\nabla V(X), \mathbb{E}\Sigma^{-1}(X - m)\rangle \\
&= \mathbb{E}\langle\nabla V(X), \Sigma^{-1}(X - m)\rangle - \langle\mathbb{E}\nabla V(X), \Sigma^{-1}(\mathbb{E}X - m)\rangle \\
&= \mathbb{E}\langle\nabla V(X), \Sigma^{-1}(X - m)\rangle.
\end{aligned}$$

Let us denote $A = \Sigma^{-1}$ and compute $\mathbb{E}\langle \nabla V(X), A(X - m)\rangle$ as follows

$$
\begin{aligned}
\mathbb{E}\langle \nabla V(X), A(X - m)\rangle &= \mathbb{E}\left( \sum_{i=1}^{d} \frac{\partial V}{\partial x_i}(X)[A(X - m)]_i \right) \\
&= \mathbb{E}\left( \sum_{i=1}^{d} \frac{\partial V}{\partial x_i}(X) \sum_{j=1}^{d} [A]_{ij}(X_j - m_j) \right) \\
&= \sum_{i=1}^{d} \sum_{j=1}^{d} [A]_{ij}\mathbb{E}\left( \frac{\partial V}{\partial x_i}(X)(X_j - m_j) \right).
\end{aligned} \tag{9}
$$

We compute $\mathbb{E}\left( \frac{\partial V}{\partial x_i}(X)(X_j - m_j) \right)$ by leveraging the following Stein's lemma (Lin et al., 2019).

**Lemma 2 (Stein's lemma)** *Let $X \sim \mathcal{N}(m, \Sigma)$ be an $d$-dimensional Gaussian random variable and $g : \mathbb{R}^d \to \mathbb{R}$ be continuously differentiable, then*

$$
\mathbb{E}\left( g(X)(X - m) \right) = \Sigma \mathbb{E}(\nabla g(X)).
$$

Applying Stein's lemma with $g = (\partial/\partial x_i)V$,

$$
\begin{aligned}
\mathbb{E}\left( \frac{\partial V}{\partial x_i}(X)(X - m) \right) &= \Sigma \mathbb{E}\left( \nabla \frac{\partial V}{\partial x_i}(X) \right) \\
&= \Sigma \mathbb{E}\left( \left[ \frac{\partial^2 V}{\partial x_1 \partial x_i}(X), \frac{\partial^2 V}{\partial x_2 \partial x_i}(X), \ldots, \frac{\partial^2 V}{\partial x_d \partial x_i}(X) \right]^\top \right).
\end{aligned}
$$

By comparing the $j$-th element of both sides, we get

$$
\mathbb{E}\left( \frac{\partial V}{\partial x_i}(X)(X_j - m_j) \right) = \sum_{k=1}^{d} \Sigma_{jk}\mathbb{E}\left( \frac{\partial^2 V}{\partial x_k \partial x_i}(X) \right).
$$

Plugging this expression into (9),

$$
\begin{aligned}
\mathbb{E}\langle \nabla V(X), A(X - m)\rangle &= \sum_{i=1}^{d} \sum_{j=1}^{d} [A]_{ij} \sum_{k=1}^{d} \Sigma_{jk}\mathbb{E}\left( \frac{\partial^2 V}{\partial x_k \partial x_i}(X) \right) \\
&= \sum_{i=1}^{d} \sum_{j=1}^{d} \sum_{k=1}^{d} [A]_{ij}\Sigma_{jk}\mathbb{E}\left( \frac{\partial^2 V}{\partial x_k \partial x_i}(X) \right) \\
&= \sum_{i=1}^{d} \sum_{k=1}^{d} \mathbb{E}\left( \frac{\partial^2 V}{\partial x_k \partial x_i}(X) \right) \sum_{j=1}^{d} [A]_{ij}\Sigma_{jk} \\
&= \sum_{i=1}^{d} \sum_{k=1}^{d} \mathbb{E}\left( \frac{\partial^2 V}{\partial x_k \partial x_i}(X) \right) [A\Sigma]_{ik} \\
&= \sum_{i=1}^{d} \sum_{k=1}^{d} \mathbb{E}\left( \frac{\partial^2 V}{\partial x_k \partial x_i}(X) \right) [I]_{ik} \\
&= \sum_{i=1}^{d} \mathbb{E}\left( \frac{\partial^2 V}{\partial x_i^2}(X) \right) \\
&= \mathrm{Tr}(\mathbb{E}\nabla^2 V(X)).
\end{aligned}
$$

Therefore,

$$
\mathcal{Q}(\mu) = 2c\,\mathrm{Tr}(\mathbb{E}\nabla^2 V(X)) - c^2\,\mathrm{Tr}(\Sigma^{-1}), \quad \text{where } X \sim \mu.
$$

## A.2 Proof of Theorem 1

Recall that Lem. 1 and the optimality condition (4) imply $\mathcal{Q}(\hat{\pi}) = c(2-c)\operatorname{Tr}(\hat{\Sigma}^{-1})$.

Now let $\mu = \mathcal{N}(m, \Sigma) \in \mathrm{BW}(\mathbb{R}^d)$, and let $(X, \hat{X})$ be the optimal coupling between $\mu$ and $\hat{\pi}$,

$$
\begin{aligned}
|\mathcal{Q}(\mu) - \mathcal{Q}(\hat{\pi})| &\leq 2c|\operatorname{Tr}(\mathbb{E}\nabla^2 V(X)) - \operatorname{Tr}(\hat{\Sigma}^{-1})| + c^2|\operatorname{Tr}(\Sigma^{-1}) - \operatorname{Tr}(\hat{\Sigma}^{-1})| \\
&= 2c|\operatorname{Tr}(\mathbb{E}\nabla^2 V(X)) - \operatorname{Tr}(\mathbb{E}\nabla^2 V(\hat{X}))| + c^2|\operatorname{Tr}(\Sigma^{-1}) - \operatorname{Tr}(\hat{\Sigma}^{-1})| \\
&\leq 2c\mathbb{E}|\operatorname{Tr}(\nabla^2 V(X)) - \operatorname{Tr}(\nabla^2 V(\hat{X}))| + c^2|\operatorname{Tr}(\Sigma^{-1}) - \operatorname{Tr}(\hat{\Sigma}^{-1})| \\
&= 2c\mathbb{E}|\Delta V(X) - \Delta V(\hat{X})| + c^2|\operatorname{Tr}(\Sigma^{-1}) - \operatorname{Tr}(\hat{\Sigma}^{-1})| \\
&\leq 2c\ell\mathbb{E}\|X - \hat{X}\| + c^2|\operatorname{Tr}(\Sigma^{-1}) - \operatorname{Tr}(\hat{\Sigma}^{-1})| \\
&\leq 2c\ell(\mathbb{E}\|X - \hat{X}\|^2)^{\frac{1}{2}} + c^2|\operatorname{Tr}(\Sigma^{-1}) - \operatorname{Tr}(\hat{\Sigma}^{-1})| \\
&= 2c\ell W_2(\mu, \hat{\pi}) + c^2|\operatorname{Tr}(\Sigma^{-1}) - \operatorname{Tr}(\hat{\Sigma}^{-1})|.
\end{aligned}
$$

Therefore

$$
\mathcal{Q}(\mu) \geq \mathcal{Q}(\hat{\pi}) - 2c\ell W_2(\mu, \hat{\pi}) - c^2|\operatorname{Tr}(\Sigma^{-1}) - \operatorname{Tr}(\hat{\Sigma}^{-1})|.
$$

So $\mathcal{Q}(\mu) > 0$ if

$$
2c\ell W_2(\mu, \hat{\pi}) + c^2|\operatorname{Tr}(\Sigma^{-1}) - \operatorname{Tr}(\hat{\Sigma}^{-1})| < \mathcal{Q}(\hat{\pi})
$$

or

$$
2\ell W_2(\mu, \hat{\pi}) + c|\operatorname{Tr}(\Sigma^{-1}) - \operatorname{Tr}(\hat{\Sigma}^{-1})| < (2-c)\operatorname{Tr}(\hat{\Sigma}^{-1}).
$$

## A.3 Proof of Theorem 2 and Remark 2

*Proof of Theorem 2*

Recall from Lem. 1: for any $\mu = \mathcal{N}(m, \Sigma) \in \mathrm{BW}(\mathbb{R}^d)$,

$$
\mathcal{Q}(\mu) = 2c\operatorname{Tr}(\mathbb{E}\nabla^2 V(X)) - c^2\operatorname{Tr}(\Sigma^{-1}), \quad \text{where } X \sim \mu.
$$

Since $V$ is $\alpha$-strongly convex, $\nabla^2 V(x) \succcurlyeq \alpha I$ for all $x \in \mathbb{R}^d$. Therefore, $\mathbb{E}\nabla^2 V(X) \succcurlyeq \alpha I$. It follows that $\operatorname{Tr}(\mathbb{E}\nabla^2 V(X)) \geq d\alpha$. Therefore, whenever $\operatorname{Tr}(\Sigma^{-1}) < (2d\alpha)/c$, $\mathcal{Q}(\mu) > 0$ and we get reduced variance.

*Proof of Remark 2*

Assuming that $V$ is convex (not necessarily strongly convex) and twice continuously differentiable. We show that: for any Gaussian random variable $X$, $\operatorname{Tr}(\mathbb{E}\nabla^2 V(X)) > 0$. By contradiction, suppose that $\operatorname{Tr}(\mathbb{E}\nabla^2 V(X)) = 0$, it follows that $\mathbb{E}\nabla^2 V(X) = 0$ since $\mathbb{E}\nabla^2 V(X)$ is symmetric and positive semidefinite. Therefore, for all $z$: $z^\top(\mathbb{E}\nabla^2 V(X))z = 0$ or $\mathbb{E}(z^\top\nabla^2 V(X)z) = 0$. Let's denote by $f$ the pdf of $X$, we have: given any fixed $z$:

$$
\int (z^\top\nabla^2 V(x)z)f(x)dx = 0
$$

Since $V$ is convex, the function under the integral is non-negative. Therefore, the integral being zero implies that the function is zero almost everywhere. The continuity of $\nabla^2 V$ further implies that the function under the integral has to be identically zero. Since $f(x) > 0$ for all $x$, we deduce $z^\top\nabla^2 V(x)z = 0$ for all $x$. Now pick $z = e_i$ where $e_i$ is the i-th basis vector, i.e., $[e_i]_j = 1$ if $j = i$, 0 otherwise, we get $\partial^2_{x_i^2}V \equiv 0$. For $i \neq j$, we set $z$ so that $z_i = z_j = 1$, $z_k = 0$ for all $k \neq i, j$, we have $\partial^2_{x_i^2}V + \partial^2_{x_j^2}V + 2\partial^2_{x_i x_j}V \equiv 0$, or $\partial^2_{x_i x_j}V \equiv 0$. We conclude that $\nabla^2 V \equiv 0$, or $V$ is an affine function. This cannot happen since $\pi \propto e^{-V}$ is a proper distribution. This contradiction implies that $\operatorname{Tr}(\mathbb{E}\nabla^2 V(X)) > 0$. Therefore, by picking $c \in (0, 2\operatorname{Tr}(\mathbb{E}\nabla^2 V(X))/\operatorname{Tr}(\Sigma^{-1}))$ we guarantee $\mathcal{Q}(\mu) > 0$.

## A.4 Discussion on Condition (7)

For convex $V$, Remark 2 guarantees that we can pick $c_k$ (that is $\mathcal{P}_k$-measurable) so that $\tau_k < 1$ almost surely. To ensure a slightly stronger condition $\|\tau_k\|_\infty < 1$ we proceed as follows. Firstly, from Lemma 1, we can set

$$\tau_k = 1 - \frac{2c_k \operatorname{Tr}(\mathbb{E}_{\mu_k} \nabla^2 V) - c_k^2 \operatorname{Tr}(\Sigma_k^{-1})}{\mathbb{E}_{\mu_k} \|\nabla V - \mathbb{E}_{\mu_k} \nabla V\|^2}.$$

On the other hand, it was shown in Diao et al. (2023, Appendix D) that we can control the eigenvalues of $\Sigma_{k+1}$ along the SGVI's iterations as follows: if $V$ is $\beta$-smooth and $\Sigma_0$ is initialized such that $\beta^{-1} I \preceq \Sigma_0$, and that the stepsize satisfies $\eta \leq \frac{1}{\beta}$, then $\beta^{-1} I \preceq \Sigma_k$ for all $k$. We note that this result also holds for SVRGVI since their computations only involve the update step of the covariance matrix. Now with $\Sigma_0 \succeq \beta^{-1} I$, it holds $\Sigma_k \succeq \beta^{-1} I$ almost surely for all $k$.

Now let us consider $\alpha > 0$ (strongly convex $V$) and pick $c_k = \frac{2\alpha d\epsilon}{\operatorname{Tr}(\Sigma_k^{-1})}$ where $\epsilon \in (0,1)$, we get

$$\tau_k \leq 1 - \frac{4\alpha^2 d^2 \epsilon(1-\epsilon)}{\operatorname{Tr}(\Sigma_k^{-1}) \mathbb{E}_{\mu_k} \|\nabla V - \mathbb{E}_{\mu_k} \nabla V\|^2}$$

$$\leq 1 - \frac{4\alpha^2 d\epsilon(1-\epsilon)}{\beta \mathbb{E}_{\mu_k} \|\nabla V - \mathbb{E}_{\mu_k} \nabla V\|^2}.$$

Therefore, with the assumption that there exists a deterministic $M_k$ such that $\mathbb{E}_{\mu_k} \|\nabla V - \mathbb{E}_{\mu_k} \nabla V\|^2 \leq M_k$ almost surely, it holds: almost surely

$$\tau_k \leq 1 - \frac{4\alpha^2 d\epsilon(1-\epsilon)}{\beta M_k}$$

which implies $\|\tau_k\|_\infty < 1$.

Now consider $\alpha = 0$ (convex but non-strongly-convex $V$) and let us pick $c_k := \frac{2\epsilon \operatorname{Tr}(\mathbb{E}_{\mu_k} \nabla^2 V)}{\operatorname{Tr}(\Sigma_k^{-1})}$ where $\epsilon \in (0,1)$, we get

$$\tau_k = 1 - \frac{4\epsilon(1-\epsilon)(\operatorname{Tr}(\mathbb{E}_{\mu_k} \nabla^2 V))^2}{\beta d M_k}.$$

If we further assume that $\|m_k\|$ and the eigenvalues of $\Sigma_k$ are also bounded above in this case, i.e., there exists a deterministic $B_k, E_k$ such that $\lambda_{\max}(\Sigma_k) \leq B_k$ and $\|m_k\| \leq E_k$ almost surely. Recall that $\Sigma_k \succeq \beta^{-1} I$.

Since $V$ cannot be affine in the entire domain, there exist $x_0 \in \mathbb{R}^d$ and $i \in \{1, 2, \ldots, d\}$ such that

$$\frac{\partial^2 V}{\partial x_i^2}(x_0) > 0.$$

By the continuity of $\partial^2 V / \partial x_i^2$, there exists $r > 0, \epsilon > 0$ such that

$$\frac{\partial^2 V}{\partial x_i^2}(x) \geq \epsilon, \quad \forall x \in B(x_0, r).$$

By denoting by $f(x; m, \Sigma)$ the pdf of $\mathcal{N}(m, \Sigma)$, we evaluate

$$\operatorname{Tr}(\mathbb{E}_{\mu_k} \nabla^2 V) \geq \int_{B(x_0, r)} \frac{\partial^2 V}{\partial x_i^2}(x) f(x; m_k, \Sigma_k) dx$$

$$\geq \epsilon (2\pi)^{-\frac{d}{2}} \det(\Sigma_k)^{-\frac{1}{2}} \int_{B(x_0, r)} e^{-\frac{1}{2}(x-m_k)^\top \Sigma_k^{-1}(x-m_k)} dx$$

$$\geq \epsilon (2\pi)^{-\frac{d}{2}} B_k^{-\frac{d}{2}} e^{-\beta E_k^2} \int_{B(x_0, r)} e^{-\beta \|x\|^2} dx.$$

A.5 PROOF OF THEOREM 3

Since Alg. 1 differs from SGVI (Diao et al., 2023) only at $\tilde{b}_k$, we will largely leverage the convergence analysis of Diao et al. (2023) but will pay extra attention to the transition of the variance reduction effect to the final bounds.

At the iteration $k$, the (deterministic) Bures–Wasserstein gradient of $\mathcal{E}_V$ at $\mu_k$ is

$$\nabla_{\text{BW}} \mathcal{E}_V(\mu_k) : x \mapsto \mathbb{E}_{\mu_k} \nabla V + (\mathbb{E}_{\mu_k} \nabla^2 V)(x - m_k)$$

and in Alg. 1 we approximate this gradient by

$$x \mapsto \tilde{b}_k + S_k(x - m_k)$$

where $\tilde{b}_k = \nabla V(X_k) - c_k \Sigma_k^{-1}(X_k - m_k)$, $S_k = \nabla^2 V(X_k)$, and $X_k \sim \mu_k$.

The error of this approximation is

$$\tilde{e}_k : x \mapsto (S_k - \mathbb{E}_{\mu_k} \nabla^2 V)(x - m_k) + (\tilde{b}_k - \mathbb{E}_{\mu_k} \nabla V).$$

Let $\mathcal{P}_k$ denote $\sigma$-algebra containing the information up to the beginning of iteration $k$, $\mathcal{P}_k = \sigma(X_0, X_1, \ldots, X_{k-1})$ for $k \in \{1, 2, \ldots, N-1\}$ and $\mathcal{P}_0$ is, by convention, the trivial $\sigma$-algebra. Let us denote

$$\tilde{\sigma}_k^2 := \mathbb{E}(\|\tilde{e}_k\|_{\mu_k}^2 | \mathcal{P}_k) = \mathbb{E}(\mathbb{E}_{x \sim \mu_k} \|(S_k - \mathbb{E}_{\mu_k} \nabla^2 V)(x - m_k) + (\tilde{b}_k - \mathbb{E}_{\mu_k} \nabla V)\|^2 | \mathcal{P}_k). \quad (10)$$

Bounding $\tilde{\sigma}_k$: we show that

$$\tilde{\sigma}_k^2 \le 3d\beta(1 + \tau_k) + 6(1 + \tau_k)\beta^3 \lambda_{\max}(\hat{\Sigma}) W_2^2(\mu_k, \hat{\pi}), \quad (11)$$

The proof of (11) is a direct extension of Diao et al. (2023, Lem. 5.6), but let us partly include it here for completeness.

First, let $\mu = \mathcal{N}(m, \Sigma)$ and $X \sim \mu$, applying Stein's lemma we get

$$\mathbb{E}\left(\frac{\partial V}{\partial x_i}(X)(X_i - m_i)\right) = \sum_{k=1}^d \Sigma_{ik} \mathbb{E}\left(\frac{\partial^2 V}{\partial x_k \partial x_i}(X)\right).$$

Summing up for $i = 1, 2, \ldots, d$

$$\sum_{i=1}^d \mathbb{E}\left(\frac{\partial V}{\partial x_i}(X)(X_i - m_i)\right) = \sum_{i=1}^d \sum_{k=1}^d \Sigma_{ik} \mathbb{E}\left(\frac{\partial^2 V}{\partial x_k \partial x_i}(X)\right),$$

which can be rewritten as

$$\mathbb{E}\langle \nabla V(X), X - m \rangle = \mathbb{E}\langle \nabla^2 V(X), \Sigma \rangle.$$

We now recall the Brascamp-Lieb inequality: let $\mu \propto \exp(-W)$ where $W$ is strictly convex and twice continuously differentiable, then

$$\text{Var}_\mu(f) \le \mathbb{E}_\mu \langle \nabla f, (\nabla^2 W)^{-1} \nabla f \rangle$$

for any smooth $f$. By using $f = (\partial/\partial x_i)V$ and $\mu = \mu_k$, we obtain

$$\text{Var}_{\mu_k}((\partial/\partial x_i)V) \le \mathbb{E}_{\mu_k}[\nabla^2 V \Sigma_k \nabla^2 V]_{ii}. \quad (12)$$

Summing (12) for $i$ from 1 to $d$

$$\mathbb{E}_{\mu_k} \|\nabla V - \mathbb{E}_{\mu_k} \nabla V\|^2 \le \text{Tr}\left(\mathbb{E}_{\mu_k}(\nabla^2 V \Sigma_k \nabla^2 V)\right) = \mathbb{E}_{\mu_k}\langle \nabla^2 V, \Sigma_k \nabla^2 V \rangle.$$

Since $X_k$ is the only source of randomness in $\tilde{e}_k$ given $\mathcal{P}_k$, the conditional expectation in (10) becomes the expectation over the randomness of $X_k$, we can write

$$\tilde{\sigma}_k^2 = \mathbb{E}\|(\nabla^2 V(X_k) - \mathbb{E}_{\mu_k} \nabla^2 V)(X - m_k) + \nabla V(X_k) - c_k \Sigma_k^{-1}(X_k - m_k) - \mathbb{E}_{\mu_k} \nabla V\|^2$$

where $X, X_k \sim \mu_k$ and $X, X_k$ are independent. We evaluate

$$
\begin{aligned}
\frac{1}{2}\tilde\sigma_k^2 &\le \mathbb{E}\|(\nabla^2 V(X_k) - \mathbb{E}_{\mu_k}\nabla^2 V)(X - m_k)\|^2 + \mathbb{E}\|\nabla V(X_k) - c_k\Sigma_k^{-1}(X_k - m_k) - \mathbb{E}_{\mu_k}\nabla V\|^2 \\
&\le \mathbb{E}((X - m_k)^\top(\nabla^2 V(X_k) - \mathbb{E}_{\mu_k}\nabla^2 V)^2(X - m_k)) + \tau_k \mathbb{E}_{\mu_k}\|\nabla V - \mathbb{E}_{\mu_k}\nabla V\|^2 \\
&= \mathbb{E}\langle(\nabla^2 V(X_k) - \mathbb{E}_{\mu_k}\nabla^2 V)^2, (X - m_k)(X - m_k)^\top\rangle + \tau_k\mathbb{E}_{\mu_k}\|\nabla V - \mathbb{E}_{\mu_k}\nabla V\|^2 \\
&= \langle\mathbb{E}_{\mu_k}(\nabla^2 V - \mathbb{E}_{\mu_k}\nabla^2 V)^2, \Sigma_k\rangle + \tau_k\mathbb{E}_{\mu_k}\|\nabla V - \mathbb{E}_{\mu_k}\nabla V\|^2 \\
&= \mathbb{E}_{\mu_k}\langle\nabla^2 V, \Sigma_k\nabla^2 V\rangle - \langle(\mathbb{E}_{\mu_k}\nabla^2 V)^2, \Sigma_k\rangle + \tau_k\mathbb{E}_{\mu_k}\|\nabla V - \mathbb{E}_{\mu_k}\nabla V\|^2 \\
&\le \mathbb{E}_{\mu_k}\langle\nabla^2 V, \Sigma_k\nabla^2 V\rangle + \tau_k\mathbb{E}_{\mu_k}\|\nabla V - \mathbb{E}_{\mu_k}\nabla V\|^2 \\
&\le (1 + \tau_k)\mathbb{E}_{\mu_k}\langle\nabla^2 V, \Sigma_k\nabla^2 V\rangle \\
&\le \beta(1 + \tau_k)\mathbb{E}_{\mu_k}\langle\nabla^2 V, \Sigma_k\rangle \\
&= \beta(1 + \tau_k)\mathbb{E}\langle\nabla V(X_k), X_k - m_k\rangle.
\end{aligned}
$$

Now by using optimal coupling between $\mu_k$ and $\hat\pi$, one can obtain (Diao et al., 2023, P.27, P.28)

$$
\mathbb{E}\langle\nabla V(X_k), X_k - m\rangle \le \frac{3d}{2} + \left(2\beta + \frac{\beta^2\operatorname{Tr}(\hat\Sigma)}{d}\right)W_2^2(\mu_k, \hat\pi)
$$

Therefore,

$$
\begin{aligned}
\tilde\sigma_k^2 &\le 3d\beta(1 + \tau_k) + (1 + \tau_k)\left(4\beta^2 + \frac{2\beta^3\operatorname{Tr}(\hat\Sigma)}{d}\right)W_2^2(\mu_k, \hat\pi) \\
&\le 3d\beta(1 + \tau_k) + 6(1 + \tau_k)\beta^3\lambda_{\max}(\hat\Sigma)W_2^2(\mu_k, \hat\pi).
\end{aligned}
$$

Bound $\mathbb{E}(\min_{k=\overline{1,N}}\mathcal{F}(\mu_k)) - \mathcal{F}(\hat\pi)$:

Lem. 5.1 in Diao et al. (2023) implies that

$$
\mathbb{E}W_2^2(\mu_{k+1}, \hat\pi) \le (1 - \alpha\eta)\mathbb{E}W_2^2(\mu_k, \hat\pi) - 2\eta(\mathbb{E}\mathcal{F}(\mu_{k+1}) - \mathcal{F}(\hat\pi)) + 2\eta^2\mathbb{E}\tilde\sigma_k^2 \tag{13}
$$

where $\alpha \ge 0$ is the strong convexity modulus of $V$.

Now using the bound (11) for $\tilde\sigma_k$,

$$
\begin{aligned}
\mathbb{E}W_2^2(\mu_{k+1}, \hat\pi) &\le (1 - \alpha\eta + 12(1 + \|\tau_k\|_\infty)\eta^2\beta^3\lambda_{\max}(\hat\Sigma))\mathbb{E}W_2^2(\mu_k, \hat\pi) \\
&\quad - 2\eta(\mathbb{E}\mathcal{F}(\mu_{k+1}) - \mathcal{F}(\hat\pi)) + 6(1 + \mathbb{E}\tau_k)\eta^2\beta d \\
&\le \exp\left(-\alpha\eta + 12(1 + \|\tau_k\|_\infty)\eta^2\beta^3\lambda_{\max}(\hat\Sigma)\right)\mathbb{E}W_2^2(\mu_k, \hat\pi) \\
&\quad - 2\eta(\mathbb{E}\mathcal{F}(\mu_{k+1}) - \mathcal{F}(\hat\pi)) + 6(1 + \mathbb{E}\tau_k)\eta^2\beta d.
\end{aligned}
$$

Therefore

$$
\begin{aligned}
2\eta(\mathbb{E}\mathcal{F}(\mu_{k+1}) - \mathcal{F}(\hat\pi)) &\le \exp\left(-\alpha\eta + 12(1 + \|\tau_k\|_\infty)\eta^2\beta^3\lambda_{\max}(\hat\Sigma)\right)\mathbb{E}W_2^2(\mu_k, \hat\pi) \\
&\quad - \mathbb{E}W_2^2(\mu_{k+1}, \hat\pi) + 6(1 + \mathbb{E}\tau_k)\eta^2\beta d
\end{aligned} \tag{14}
$$

Since we are considering the convex case, set $\alpha = 0$ and denote $C_k = 12(1 + \|\tau_k\|_\infty)\beta^3\lambda_{\max}(\hat\Sigma)$ and $D_{-1} = 0$, $D_k = -C_0 - C_1 - \ldots - C_k$ for $k = 0, 1, \ldots, N-1$. By definition, $D_k + C_k = D_{k-1}$ for all $k = 0, 1, \ldots, N-1$. Rewrite (14) as

$$
2\eta(\mathbb{E}\mathcal{F}(\mu_{k+1}) - \mathcal{F}(\hat\pi)) \le \exp\left(C_k\eta^2\right)\mathbb{E}W_2^2(\mu_k, \hat\pi) - \mathbb{E}W_2^2(\mu_{k+1}, \hat\pi) + 6(1 + \mathbb{E}\tau_k)\eta^2\beta d.
$$

Multiply both sides with $\exp(D_k\eta^2)$ we get

$$
\begin{aligned}
&2\eta\exp(D_k\eta^2)(\mathbb{E}\mathcal{F}(\mu_{k+1}) - \mathcal{F}(\hat\pi)) \\
&\le \exp\left((D_k + C_k)\eta^2\right)\mathbb{E}W_2^2(\mu_k, \hat\pi) - \exp(D_k\eta^2)\mathbb{E}W_2^2(\mu_{k+1}, \hat\pi) + 6(1 + \mathbb{E}\tau_k)\eta^2\beta d\exp(D_k\eta^2)
\end{aligned}
$$

and, by using the backward recursion $D_k + C_k = D_{k-1}$, can be rewritten as

$$2\eta \exp(D_k\eta^2)(\mathbb{E}\mathcal{F}(\mu_{k+1}) - \mathcal{F}(\hat{\pi}))$$
$$\leq \exp\left(D_{k-1}\eta^2\right)\mathbb{E}W_2^2(\mu_k, \hat{\pi}) - \exp(D_k\eta^2)\mathbb{E}W_2^2(\mu_{k+1}, \hat{\pi}) + 6(1 + \mathbb{E}\tau_k)\eta^2\beta d \exp(D_k\eta^2)$$

Telescope for $k$ from $0$ to $N-1$

$$2\eta \sum_{k=0}^{N-1} \exp(D_k\eta^2)(\mathbb{E}\mathcal{F}(\mu_{k+1}) - \mathcal{F}(\hat{\pi}))$$

$$\leq W_2^2(\mu_0, \hat{\pi}) - \exp(D_{N-1}\eta^2)\mathbb{E}W_2^2(\mu_N, \hat{\pi}) + 6\eta^2\beta d \sum_{k=0}^{N-1} (1 + \mathbb{E}\tau_k)\exp(D_k\eta^2)$$

$$\leq W_2^2(\mu_0, \hat{\pi}) + 6\eta^2\beta d \sum_{k=0}^{N-1} (1 + \mathbb{E}\tau_k)\exp(D_k\eta^2).$$

We see that

$$D_k = -\frac{C}{2}\left(k + 1 + \sum_{i=0}^{k} \|\tau_i\|_\infty\right)$$

where $C = 24\beta^3\lambda_{\max}(\hat{\Sigma})$.

Let us denote $\tilde{S}_N(\eta) = \sum_{k=0}^{N-1} \exp(D_k\eta^2)$. It holds

$$\mathbb{E}\left(\min_{k=1,2,\ldots,N} \mathcal{F}(\mu_k)\right) - \mathcal{F}(\hat{\pi}) \leq \frac{W_2^2(\mu_0, \hat{\pi})}{2\eta\tilde{S}_N(\eta)} + 3\eta\beta d \sum_{k=0}^{N-1} (1 + \mathbb{E}\tau_k)\frac{\exp(D_k\eta^2)}{\tilde{S}_N(\eta)}.$$

It holds

$$\sum_{k=0}^{N-1} (1 + \mathbb{E}\tau_k)\frac{\exp(D_k\eta^2)}{\tilde{S}_N(\eta)} \leq 1 + \tau_{\max,E} \tag{15}$$

and

$$\tilde{S}_N(\eta) = \sum_{k=0}^{N-1} \exp(D_k\eta^2)$$
$$= \sum_{k=0}^{N-1} \exp\left(-\frac{C}{2}(k + 1 + \sum_{i=0}^{k} \|\tau_i\|_\infty)\eta^2\right)$$
$$\geq \sum_{k=0}^{N-1} \exp\left(-\frac{C}{2}(k + 1 + (k+1)\tau_{\max,\infty})\eta^2\right)$$
$$= \sum_{k=0}^{N-1} \left[\exp\left(-C(k+1)\eta^2\right)\right]^{\frac{\tau_{\max,\infty}+1}{2}}.$$

On the other hand, for any $b > 0$, the function $f(s) = b^s$ is convex. By tangent inequality $f(s) \geq f(1) + f'(1)(s - 1)$, we get

$$b^s \geq b + b\ln(b)(s - 1). \tag{16}$$

Applying the inequality (16) with $b = \exp\left(-C(k+1)\eta^2\right)$ and $s = (\tau_{\max,\infty} + 1)/2$

$$\left[\exp\left(-C(k+1)\eta^2\right)\right]^{\frac{\tau_{\max,\infty}+1}{2}} \geq \exp\left(-C(k+1)\eta^2\right) + C(k+1)\eta^2 \exp\left(-C(k+1)\eta^2\right)\left(\frac{1 - \tau_{\max,\infty}}{2}\right)$$

$$= \exp\left(-C(k+1)\eta^2\right)\left(1 + C\eta^2(k+1)\left(\frac{1 - \tau_{\max,\infty}}{2}\right)\right)$$

$$\geq \exp\left(-C(k+1)\eta^2\right)\left(1 + C\eta^2\left(\frac{1 - \tau_{\max,\infty}}{2}\right)\right).$$

Therefore,

$$
\tilde{S}_N(\eta) \geq \left(1 + \frac{C\eta^2(1 - \tau_{\max,\infty})}{2}\right) \sum_{k=1}^{N} \exp(-Ck\eta^2)
$$

$$
\geq \left(1 + \frac{C\eta^2(1 - \tau_{\max,\infty})}{2}\right) \sum_{k=1}^{\min\{N, \lfloor(C\eta^2)^{-1}\rfloor\}} \exp(-Ck\eta^2)
$$

$$
\geq \left(1 + \frac{C\eta^2(1 - \tau_{\max,\infty})}{2}\right) \sum_{k=1}^{\min\{N, \lfloor(C\eta^2)^{-1}\rfloor\}} \frac{1}{e}
$$

$$
= \frac{1}{e}\left(1 + \frac{C\eta^2(1 - \tau_{\max,\infty})}{2}\right) \min\{N, \lfloor(C\eta^2)^{-1}\rfloor\}.
$$

By using the basic inequality $1/\min(a, b) \leq 1/a + 1/b$, we get

$$
\frac{1}{\tilde{S}_N(\eta)} \leq \frac{e}{1 + \dfrac{C\eta^2(1 - \tau_{\max,\infty})}{2}} \left(\frac{1}{N} + \frac{1}{\lfloor(C\eta^2)^{-1}\rfloor}\right)
$$

$$
\lesssim \frac{e}{1 + \dfrac{C\eta^2(1 - \tau_{\max,\infty})}{2}} \left(\frac{1}{N} + C\eta^2\right)
$$

asymptotically at small $\eta > 0$.

Therefore,

$$
\mathbb{E}\left(\min_{k=1,2,\dots,N} \mathcal{F}(\mu_k)\right) - \mathcal{F}(\hat{\pi}) \leq \frac{e}{1 + \dfrac{C\eta^2(1 - \tau_{\max,\infty})}{2}} \left(\frac{1}{2\eta N} + \frac{C\eta}{2}\right) W_2^2(\mu_0, \hat{\pi}) + 3\eta\beta d(1 + \tau_{\max,E}).
$$

### A.6 PROOF OF THEOREM 4

Since $V$ is $\alpha$-strongly convex with $\alpha > 0$, $\mathbb{E}_{\hat{\pi}}(\nabla^2 V) \succcurlyeq \alpha I$, so $\lambda_{\min}(\mathbb{E}_{\hat{\pi}}(\nabla^2 V)) \geq \alpha$.

It follows that

$$
\lambda_{\max}(\hat{\Sigma}) = \frac{1}{\lambda_{\min}(\hat{\Sigma}^{-1})} = \frac{1}{\lambda_{\min}(\mathbb{E}_{\hat{\pi}}(\nabla^2 V))} \leq \frac{1}{\alpha}.
$$

Using this inequality in the bound for $\tilde{\sigma}_k$ in (11), we get

$$
\tilde{\sigma}_k^2 \leq 3d\beta(1 + \tau_k) + \frac{6(1 + \tau_k)\beta^3}{\alpha} W_2^2(\mu_k, \hat{\pi}).
$$

Using this bound for (13),

$$
\mathbb{E}W_2^2(\mu_{k+1}, \hat{\pi}) \leq (1 - \alpha\eta)\mathbb{E}W_2^2(\mu_k, \hat{\pi}) - 2\eta(\mathbb{E}\mathcal{F}(\mu_{k+1}) - \mathcal{F}(\hat{\pi}))
$$

$$
+ 2\eta^2 \mathbb{E}\left(3d\beta(1 + \tau_k) + \frac{6(1 + \tau_k)\beta^3}{\alpha} W_2^2(\mu_k, \hat{\pi})\right)
$$

$$
= \left(1 - \alpha\eta + \frac{12(1 + \|\tau_k\|_\infty)\eta^2\beta^3}{\alpha}\right) \mathbb{E}W_2^2(\mu_k, \hat{\pi}) + 6d\beta\eta^2(1 + \mathbb{E}\tau_k)
$$

$$
\leq \exp\left(-\alpha\eta + \frac{12(1 + \|\tau_k\|_\infty)\eta^2\beta^3}{\alpha}\right) \mathbb{E}W_2^2(\mu_k, \hat{\pi}) + 6d\beta\eta^2(1 + \mathbb{E}\tau_k).
$$

Now with $\eta \leq \alpha^2/(48\beta^3)$,

$$
\frac{12(1 + \|\tau_k\|_\infty)\eta^2\beta^3}{\alpha} \leq \frac{(1 + \|\tau_k\|_\infty)\eta\alpha}{4}.
$$

Therefore

$$\mathbb{E}W_2^2(\mu_{k+1}, \hat{\pi}) \leq \exp\left(\left(\frac{-3 + \|\tau_k\|_\infty}{4}\right)\eta\alpha\right)\mathbb{E}W_2^2(\mu_k, \hat{\pi}) + 6d\beta\eta^2(1 + \mathbb{E}\tau_k)$$

$$\leq \exp\left(\left(\frac{-3 + \tau_{\max,\infty}}{4}\right)\eta\alpha\right)\mathbb{E}W_2^2(\mu_k, \hat{\pi}) + 6d\beta\eta^2(1 + \tau_{\max,E}).$$

Telescope this inequality, we get

$$\mathbb{E}W_2^2(\mu_N, \hat{\pi}) \leq \exp\left(-N\left(\frac{3 - \tau_{\max,\infty}}{4}\right)\eta\alpha\right)W_2^2(\mu_0, \hat{\pi}) + \frac{6(1 + \tau_{\max,E})\eta^2\beta d}{1 - \exp\left(-\frac{(3 - \tau_{\max,\infty})\eta\alpha}{4}\right)}$$

$$\lesssim \exp\left(-\frac{N(3 - \tau_{\max,\infty})\eta\alpha}{4}\right)W_2^2(\mu_0, \hat{\pi}) + \frac{24(1 + \tau_{\max,E})\beta\eta d}{(3 - \tau_{\max,\infty})\alpha}$$

asymptotically at small $\eta > 0$.

## B  ADDITIONAL EXPERIMENTAL DETAILS

### B.1  LAPLACE APPROXIMATION

Laplace approximation fits a Gaussian approximation by finding the mode of the target (MAP estimate for infernece) and forming a second order approximation at that point. The approximation is given by

$$\mathcal{N}\left(x_{\text{MAP}}, \left(\nabla^2 V(x_{\text{MAP}})\right)^{-1}\right).$$

We use BFGS optimizer (Nocedal & Wright, 2006) as implemented in SciPy (Virtanen et al., 2020) to find the (numerical) MAP estimate, and form the approximation according to the local curvature around the point. Convergence of the estimate was validated manually.

### B.2  VARIATIONAL INFERENCE IN THE EUCLIDEAN GEOMETRY

The baseline method EVI optimizes for the approximation over its parameters $m$ and $\Sigma$ in the Euclidean geometry of the parameter space, using Cholesky factorization for parameterizing the covariance. This is done by maximizing the Evidence Lower BOund (ELBO)

$$\mathcal{L}(m, \Sigma) = \mathbb{E}_{q_{m,\Sigma}(z)}\left[\log p(x, z) - \log q_{m,\Sigma}(z)\right],$$

which is equivalent to minimizing the KL divergence. We use single-sample reparameterization estimates for the gradient. Furthermore, by stopping the gradient after sampling $z$, we remove the Fisher score from the gradient computation, giving an unbiased estimator of the gradient of the ELBO with potentially lower variance (Roeder et al., 2017). We use Adam (Kingma & Ba, 2015) optimizer and the learning rates and number of iterations found in Table 1, found to achieve good convergence. Our implementation is based on the code provided by Modi et al. (2024).

| Experiment | Dimension | Learning Rate | Iterations |
|---|---|---|---|
| Gaussian | 10 | 0.01 | 5,000 |
| Gaussian | 50 | 0.01 | 5,000 |
| Gaussian | 200 | 0.001 | 10,000 |
| Student-t | 200 | 0.001 | 8,000 |
| Logistic Regression | 200 | 0.01 | 3,000 |

Table 1: Optimization details for EVI.

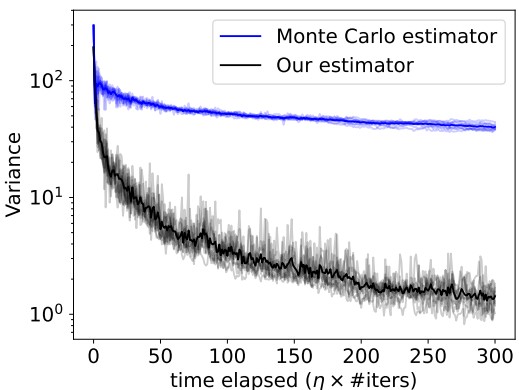

Figure 6: Student's t experiment: variance along iterations

### B.3 STUDENT'S T DISTRIBUTION

Consider a $d$-dimensional Student-t distribution with location $\mu$, scale matrix $\Sigma$ and $\nu$ degrees of freedom. Its negative log density (up to a constant), gradient and Hessian are given by:

$$V(x) = \frac{1}{2}(\nu + d) \log\left(1 + \frac{1}{\nu}(x - \mu)^\top \Sigma^{-1}(x - \mu)\right),$$

$$\nabla V(x) = \frac{(\nu + d)}{\nu + (x - \mu)^\top \Sigma^{-1}(x - \mu)} \Sigma^{-1}(x - \mu),$$

$$\nabla^2 V(x) = \frac{\nu + d}{\nu + (x - \mu)^\top \Sigma^{-1}(x - \mu)} \Sigma^{-1} - \frac{2(\nu + d)}{(\nu + (x - \mu)^\top \Sigma^{-1}(x - \mu))^2} \Sigma^{-1}(x - \mu)(x - \mu)^\top \Sigma^{-1}.$$

### B.4 VARIANCE ALONG ITERATIONS

We further report in Figures 5 and 6 the variance of our proposed estimator and the Monte Carlo estimator along the SVRGVI's iterates. The variance is computed empirically using 5000 i.i.d. samples at each iteration. The results demonstrate that our estimator consistently achieves a significantly smaller variance compared to the Monte Carlo estimator.

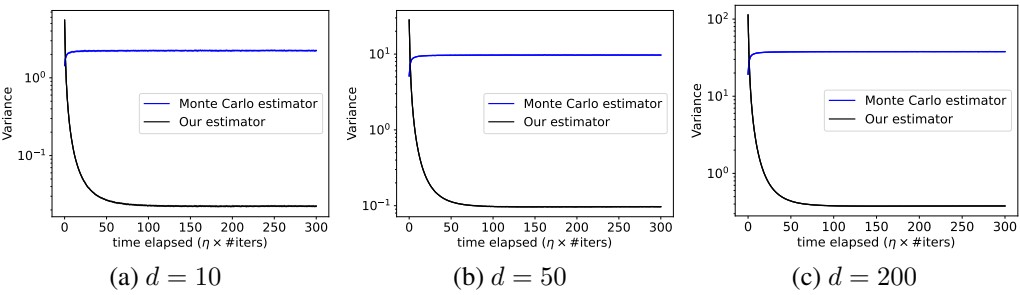

(a) $d = 10$        (b) $d = 50$        (c) $d = 200$

Figure 5: Gaussian experiment: variance along iterations.

### B.5 COMPARISONS AGAINST THE MINIBATCH APPROACH

A straightforward approach to reduce the variance is to use more MC samples per iteration. In this experiment, we use $m$ samples for SGVI at each iteration, where $m \in \{1, 10, 100\}$. Fig. 7 illustrates that SGVI requires approximately 100 samples per iteration to achieve a performance level comparable to SVRGVI using only one sample per iteration.

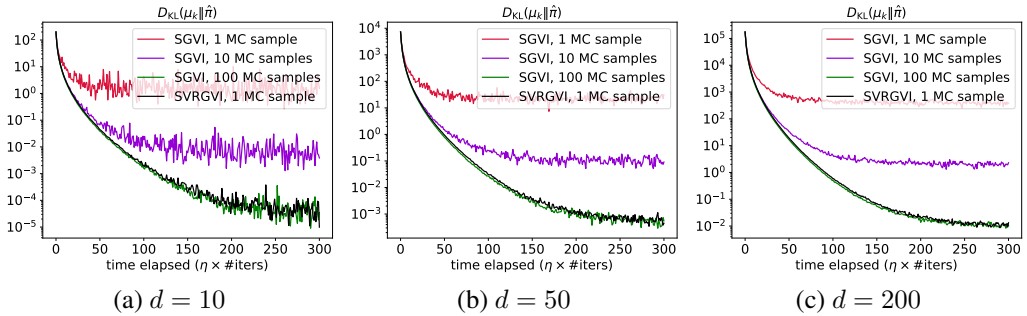

Figure 7: Minibatch-SGVI with MC samples versus SVRGVI in the Gaussian experiment

In Euclidean VI, Buchholz et al. (2018) showed that quasi-MC samples can result in a better estimator (with smaller variances) than standard MC samples. Since the idea is universal, we can use quasi-MC samples to improve the performance of SGVI as well. Fig. 8 confirms that using quasi-MC samples indeed leads to better performance in practice [3]. SGVI now needs around $50$ quasi-MC samples to reach our performance, and with $100$ quasi-MC samples, SGVI surpasses our performance.

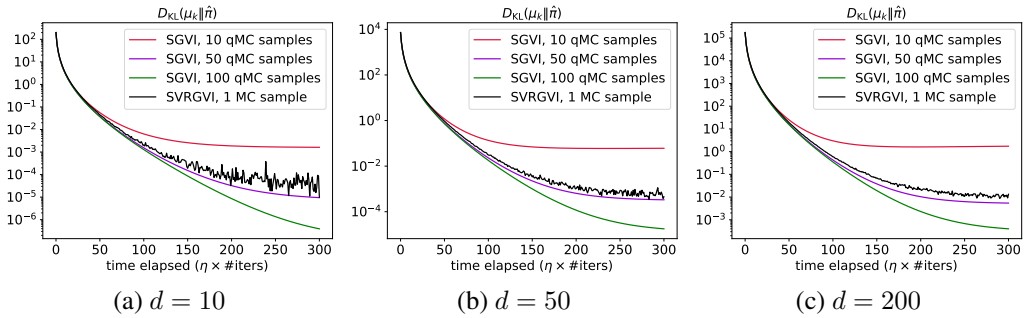

Figure 8: Minibatch-SGVI with quasi-MC samples versus SVRGVI in the Gaussian experiment

### B.6 EFFECT OF $c$

In this experiment, we study the impact of $c$ on the performance of SVRGVI. In Figures 9, 10, 11, we report the performance of SVRGVI in the Gaussian, Student's, and Bayesian logistic regression experiments when $c$ varies in $\{0.0, 0.5, 0.8, 1.0, 1.2, 1.5, 2.0\}$. The results indicate that performance improves as $c$ increases from $0$ to $1$, peaking around $c = 1$, and then degrades as $c$ continues increasing to $2.0$. Furthermore, the performance is somewhat symmetric around $c = 1$, e.g., $c = 0.8$ and $c = 1.2$ yield similar results. We therefore confirm that $c$ being around $1$ works best in practice.

### B.7 EFFECT OF STEP SIZE

We conduct an experiment to compare the performances of different algorithms with varying step sizes. We consider Gaussian targets with $D = 100$. We fix the number of steps to $300$, and vary the step size between $[0.125, 0.25, 0.5, 1.0]$. The results, as shown in Figure 12, indicates that while the previous method requires a relatively small step size to work relatively well, our algorithm is able to work robustly with large step sizes and achieves the best performances under all step sizes.

---

[3]We exclude the case of using a single quasi-MC sample, as it coincides with the mean of the Gaussian variational distribution when employing a Scrambled Sobol sequence. In this specific experiment, this leads to an optimal—but misleading—result purely by coincidence.

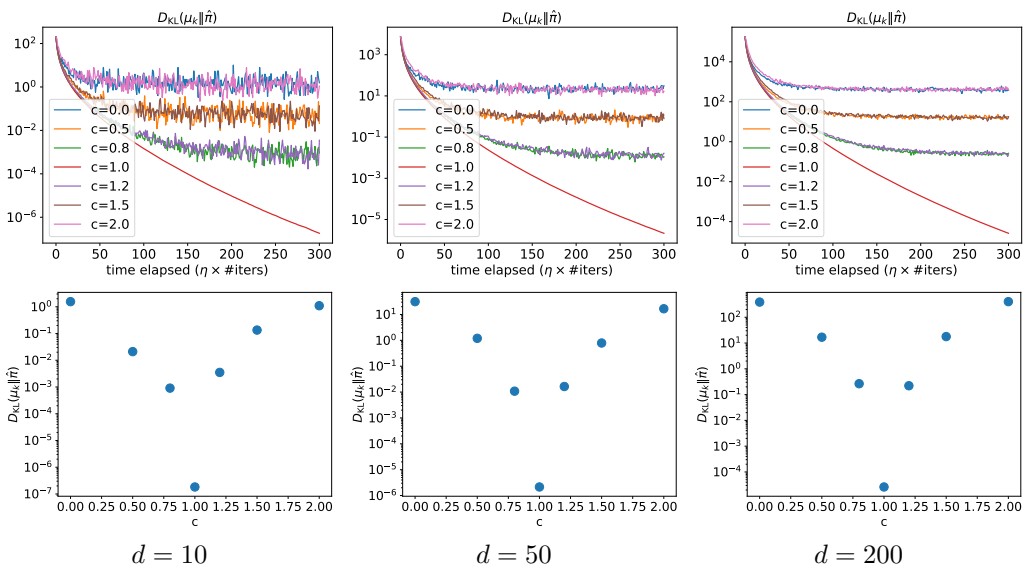

Figure 9: Gaussian experiments: **Upper row.** KL divergence along iterations; **Lower row:** final KL divergence.

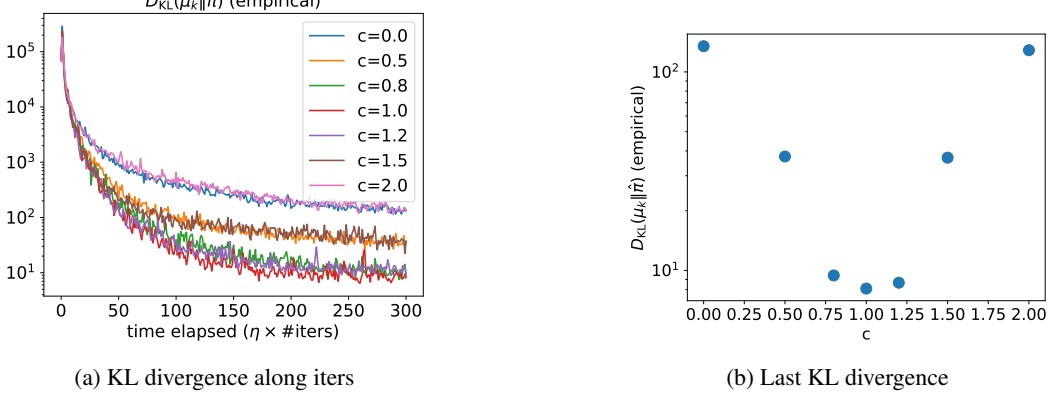

(a) KL divergence along iters

(b) Last KL divergence

Figure 10: Student's t experiment

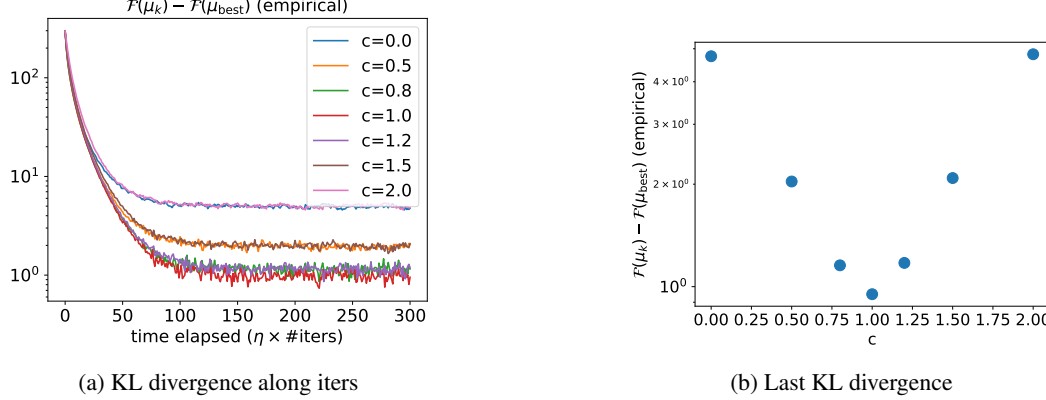

(a) KL divergence along iters

(b) Last KL divergence

Figure 11: Bayesian linear regression experiment

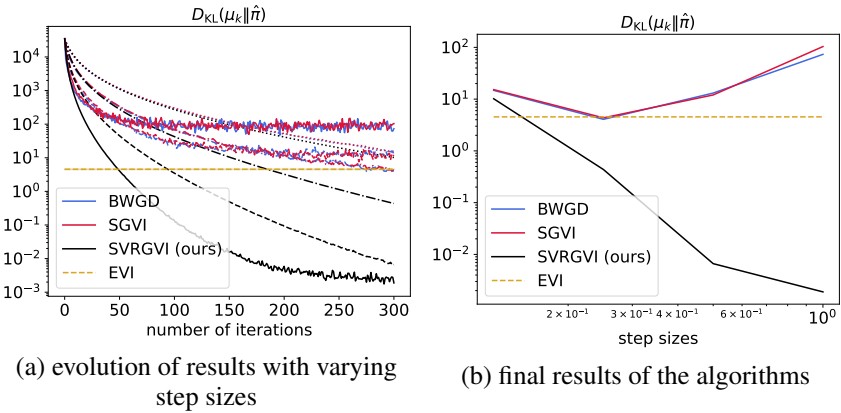

(a) evolution of results with varying step sizes

(b) final results of the algorithms

Figure 12: Results of different algorithms with varying step sizes in the Gaussian experiment

