# OpenReview forum: "Stochastic variance-reduced Gaussian variational inference on the Bures-Wasserstein manifold"
_ICLR.cc/2025/Conference — ICLR 2025 Poster_

### Official Review · Reviewer_NBYM · 2024-10-28

**Soundness:** 2
**Presentation:** 3
**Contribution:** 2
**Rating:** 5
**Confidence:** 4

**Summary:**

This paper considers the variational inference (VI) problem by minimizing the Kullback-Leibler (KL) divergence over the Bures-Wasserstei space of Gaussian distributions. Building on the work by Diao et al., this paper employs the forward-backward Euler method to tackle the composite nature of the KL objective and non-smoothness of the entropy. The forward step involves computing the Bures-Wasserstein (BW) gradient, which needs to be estimated using Monte Carlo (MC) method with a single sample. However, this approach results in noisy BW gradients, leading to inefficient convergence. The main contribution of this paper is to overcome this limitation, proposing a variance-reduced estimator for the BW gradient, leveraging a control variate approach. In addition, this paper also provides convergence analysis and present experimental results that demonstrate the effectiveness of the proposed method.

**Strengths:**

The paper is well structured, clearly presenting the problem and the proposed solution. The main idea of using a control variate to reduce the variance of BW gradient is both well-motivated and reasonable. Additionally, it provides strong theoretical foundations for the derived variance-reduced Gaussian VI. The experimental results further validate the effectiveness of the proposed method.

**Weaknesses:**

I have two major concerns regarding the proposed method in this paper:

1) A straightforward approach to addressing the noisy BW gradient is to sample multiple $X_{k}$ from $\mu_{k}$. This is both effective and efficient, requiring low extra computational cost if estimating $\nabla V$ and $\nabla^{2}V$ is not expensive. I am curious how the performance of the proposed method compares to this trivial approach.

2) The paper only considers the case where $\mu_{k}$ is Gaussian, and the proposed method appears to work well for simple target distributions. Note that both compared methods (BWGD and SGVI) can be extended for Gauss mixtures, while extending the proposed method in this paper to cases of Gaussian mixtures may not trivial. In such cases, the proposed control variate approach might lose its effectiveness and efficiency, as $E[Z_{k}]$ is not zero (the mean) and estimating it is non-trivial.

**Questions:**

Please see the weakness section.

---

> ### Author Response · Authors · 2024-11-20
>
> We are thankful to the reviewer for the positive comments on our work.
>
> # Weaknesses
>
> 1. Using more samples indeed reduces the variances, but is far less efficient. Our method is effectively as easy to implement and has virtually no computational overhead. We now added a new experiment where we empirically compare against the naive approach to make the difference clear. Figure 7 in Appendix B.5 shows that to reach the same convergence rate one needs to use around 100 samples (and hence gradient oracles) per iteration, compared to just **one sample** for our method.
>
> 2. To clarify, while BWGD can be extended for Gaussian mixture approximations, to our knowledge, SGVI cannot. We build on SGVI and retain the same limitations. The reason, we think, is quite understandable: the whole point of SGVI is the tractability of the backward step (JKO step), which is known to have a closed-formed solution for the family of single Gaussians. It is still unclear whether this step is still tractable for Gaussian mixtures.

---

> > ### Author Response · Authors · 2024-11-24
> >
> > Dear Reviewer NBYM,
> >
> > As the discussion phase is ending very soon, please read our rebuttal and update the score if the rebuttal did address your concerns.
> >
> > Best regards,
> > The Authors :)

---

> > > ### Comment · Reviewer_NBYM · 2024-11-24
> > > **Thank you for your clarification**
> > >
> > > Regarding the first concern, I appreciate the authors’ efforts to demonstrate that the MC samples are less efficient compared to the proposed control variate.
> > >
> > > For the second concern, similar to SGVI, the proposed method is limited to Gaussian distributions and cannot be extended to other distributions. This limitation arises because the backward step is only tractable when optimizing over the space of a single Gaussian. As a result, both SGVI and the proposed method struggle to approximate more complex target distributions. In contrast, methods like BWGD, which can be extended to Gaussian mixtures, are better to handle such targets. Therefore, the applicability of the proposed method is somewhat limited. Given this, I would like to maintain my original score.

---

> ### Author Response · Authors · 2024-11-24
> **Thank you for your reply**
>
> Thank you for the reply.
>
> We further clarify that the difference between BWGD and SGVI is: that BWGD is forward Euler, while SGVI is forward-backward Euler. Although we focus on SGVI (because the entropy is non-smooth, making SGVI the method of choice in general), we can use the same idea to improve the Bures-Wasserstein gradient used in BWGD for a single Gaussian case (because the BW gradient used in BWGD is the BW gradient used in the forward step of SGVI plus a computable deterministic term).
>
> Additionally, we clarify that: different from the single Gaussian case, BWGD for **mixture of Gaussians** is a **deterministic method** -- using **quadrature based on sigma points** to compute expectations. Therefore, developing variance reduction for a deterministic method is unnecessary.

---

### Official Review · Reviewer_7JyP · 2024-11-01

**Soundness:** 4
**Presentation:** 4
**Contribution:** 3
**Rating:** 6
**Confidence:** 4

**Summary:**

The paper propose improvements on a gaussian variational inference algorithm in the Bures-Wasserstein manifold.  The idea is to find the best Gaussian approximation (in the KL sense) to a probability distribution of the form $Ce^{-V}$ (where $C$ is unknonw) and to use the Wasserstein geometry on the space of Gaussian distribution (Bures-Wasserstein manifold) to guide us toward the optimal solution.  Several authors have studied this question in the last few year leading to an implicit-explicit algorithm.  In this framework the implicit backward (JKO-style) step is computable explicitly and the explicit forward step involves the computation of the the expectations of the gradient and  the Laplacian of $V$ under gaussian measures.  The central contribution of the paper is to perform a variance reduction (using control variates)  at each step and to study the effect of this on the rate of convergence of the algorithm.  Some synthetic examples are studied.

**Strengths:**

I found the paper to be very overall very well-written with a very clear presentation of the background, literature, and of the technical ideas involved.  The results and the proofs are clearly presented.  Since the measure involved are Gaussian, the analysis of the control variate is quite straightforward and this lead to an improved algorithm at essentially no additional cost.  I think that it is the main contribution of the paper to point out this very cheap improvement to the algorithm. The empirical results also supports this improvement of the stability of the algorithm leading to a substantially better approximation.  At the technical level, control variates for Gaussian measures are quite straightforward and the analysis of the variance reduction relies on standard techniques as well (Stein's lemma).  The convergence of the algorithms is then analyzed by leveraging the results recently obtained Diao et a. (ICML 23) and propagating the improved variance through the convergence analysis of Diao et al.  This leads to improved convergence results.

**Weaknesses:**

I think the main weakness is the lack of interesting examples. The algorithms seems to do great on the synthetic examples but it remains to be seen how it will perform in a realistic  Bayesian inference example based on real data.

**Questions:**

+  Are there any obstacles in applying this to more realistic statistical inference problems?  I really do appreciate the mathematical elegance  of the proposed method but t would be nice to have a better test problem  to make the case for the method.

+ Any head-to-head comparisons with different inferences methods such that MCM, Hamiltonian MC, or likelihood free inferences (e.g. using diffusions models) on a test problem would be helpful to convince the reader of the efficiency of the method.,

+  The improvement due to control variates looks quite impressive (as it  sometimes does with control variates!).  The convergence analysis  is nice but it is not clear it shed light on this improvement. Any other explanation, even the hand-waving kind?

---

> ### Author Response · Authors · 2024-11-20
>
> We are thankful to the reviewer for the positive comments on our work.
>
> # Questions
>
> - In principle, there are no obstacles to running for realistic statistical inference problems because the method is universal -- as long as we can access the gradient and the Hessian of the log-target. Our experimentation focused on synthetic problems as in previous BW-VI works to best illustrate and quantify the effect of the variance reduction. We leave empirical comparisons on more challenging tasks as future work; they need more work on careful tuning of learning rates etc. that could not be fit in this paper.
>
> - While comparisons across different families of inference methods (here VI vs MCMC) are interesting, they are best left for papers where that is the main objective contribution (for example, Yao et al. "Yes, but Did It Work?: Evaluating Variational Inference", ICML 2018). Already defining the metrics for such a comparison would be outside of our scope. Both our theoretical analysis and empirical results are on convergence rates and gradient variances, which are meaningful quantities for VI formulated as an optimization problem, but not for MCMC. Accuracy comparisons, in turn, would mostly depend on the experimental setup (e.g. whether the approximation family contains the true posterior or not) and not so much on the specific VI algorithm.
>
> - We now empirically quantify the variance reduction for the gradients in Appendix B.5, to show that the improved convergence is because of the variance reduction. One possible explanation is that the deterministic forward-backward (FB) Euler using BW geometry works very well, hence closely following the trajectory of FB Euler would be optimal. We also note that the improvement is more pronounced in high dimensions. Comparing the bounds of SGVI and SVRGVI, variance reduction helps scale down the term relating to the dimension $d$, where a small scaling factor can make a big difference when $d$ is large.

---

> > ### Comment · Reviewer_7JyP · 2024-11-22
> >
> > Thank you for your answers. I leave my score unchanged.

---

### Official Review · Reviewer_qVPq · 2024-11-02

**Soundness:** 2
**Presentation:** 2
**Contribution:** 2
**Rating:** 6
**Confidence:** 4

**Summary:**

- The authors propose a control-variate-based variance reduction technique for gradient estimation in stochastic Gaussian variational inference (SGVI).
- They theoretically demonstrate that, in specific cases such as high-curvature distributions, the gradient estimator with their method achieves lower variance than the standard SGVI.
- Additionally, they show that their variance reduction scheme enhances the convergence of SGVI when $V(x)$ is (strongly) convex.
- Empirically, they confirmed that their method improves the convergence properties of SGVI.

**Strengths:**

- The authors extend the control variate method—a well-established variance reduction technique in stochastic variational inference (VI)—to variational inference in Bures-Wasserstein space.
- The variance reduction performance of the proposed method is theoretically guaranteed in (strongly) convex settings.
- The authors demonstrate improved convergence over existing SGVI and Bures-Wasserstein gradient descent (BWGD) methods through several numerical experiments.

**Weaknesses:**

I would like to express my sincere respect for the efforts the authors have invested in this paper. However, I am unable to strongly recommend it for acceptance at ICLR 2025 for the following reasons:

### Concerns Regarding Theoretical Analyses:
- The evaluation of the derived bound is primarily qualitative and limited to a comparison with the existing upper bounds in [Diao et al., 2023]. However, detailed analysis of each term and the behavior of the bound as the iteration count $N$ approaches infinity—specifically in terms of convergence and the conditions required for convergence—is somewhat lacking. This limits a comprehensive understanding of the proposed method’s advantages and challenges the assessment of this paper’s contribution.
- In the convex case, the variance reduction effect is guaranteed only in the neighborhood of the optimal solution (Thm. 1). However, the convergence analysis in Thm. 3 assumes that variance reduction holds “throughout all iterations of the algorithm,” which seems to suggest that parameters are already near the optimal solution from the outset—a somewhat unrealistic assumption (or am I misinterpreting this?).
- Additionally, from the proof, it appears that the improvement in the bound is only valid under the aforementioned assumption: “when variance reduction is assured for all iterations.” This assumption seems to diverge from the result in Thm. 1. Is there a clear link between them? Is the radius $r$ of this region always sufficiently small?

### Concerns Regarding Numerical Experiments:
- One key motivation for variance reduction in stochastic variational inference is to enhance convergence and generalization in real-world data applications (e.g., [Ruiz et al. (2016); Roeder et al. (2017); Miller et al. (2017); Buckholz et al. (2018)]). However, the numerical experiments in this paper are primarily conducted on synthetic data, and the discussion of the practical applicability of the proposed method is somewhat limited.
- Additionally, there are no direct measurements of gradient variance, leaving insufficient empirical evidence regarding the extent of gradient variance reduction compared to existing methods. If the empirically measured gradient variance does not differ substantially from that of other methods, it is possible that other factors are driving the observed improvement in convergence. Furthermore, based on the results presented, the effectiveness of gradient variance reduction appears to depend on the setting of $c$. Practical guidance on choosing $c$, based on an analysis of the sensitivity of variance-reduction performance to $c$, would be beneficial. Currently, such guidance is absent, leaving limited information on how to apply the proposed method in practice.
- When comparing with variance-reduction methods in SVI in Euclidean space (EVI), this study adopts the method of Roeder et al. (2017); however, the rationale for this choice is unclear. While both methods share a similarity in achieving performance improvements through modest algorithmic modifications, concluding that the proposed method outperforms EVI without comparison to more recent approaches seems premature (e.g., [Miller et al. (2017); Buckholz et al. (2018)]).

### Lack of Discussion on Other Variance Reduction Studies in EVI:
- Recent years have seen substantial progress in variance reduction research within EVI, with many theoretical analyses and methods proposed (e.g., [Kim et al. (2023); Domke et al. (2023)]). These approaches extend beyond control variate methods to include enhancements to Monte Carlo methods and techniques such as reparameterization tricks in Gaussian settings, as in Roeder et al. (2017). While the use of control variates as an initial approach to variance reduction in SGVI is reasonable, the motivation for extending only the control variate approach may seem somewhat limited given the variety of variance reduction techniques available. It might be helpful to mention, for example, the specific challenges or limitations that prevent the application of alternative variance reduction techniques in SGVI, if applicable.

### Citation:
- Ruiz et al. (2016): Francisco J. R. Ruiz, Michalis K. Titsias, and David M. Blei. The Generalized Reparameterization Gradient. NeurIPS2016. https://arxiv.org/abs/1610.02287.
- Miller et al. (2017): Andrew C. Miller, Nicholas J. Foti, Alexander D'Amour, and Ryan P. Adams. Reducing Reparameterization Gradient Variance. NeurIPS2017. https://arxiv.org/abs/1705.07880
- Buckholz et al. (2018): A. Buchholz, F. Wenzel, and S. Mandt. Quasi-Monte Carlo Variational Inference. ICML2018. https://arxiv.org/abs/1807.01604.
- Kim et al. (2023): K. Kim, K. Wu, J. Oh, and J. R. Gardner. Practical and Matching Gradient Variance Bounds for Black-Box Variational Bayesian Inference. ICML2023. https://proceedings.mlr.press/v202/kim23w.html.
- Domke et al. (2023): J. Domke, R. Gower, and G. Garrigos. Provable convergence guarantees for black-box variational inference. NeurIPS2023. https://proceedings.neurips.cc/paper_files/paper/2023/hash/d0bcff6425bbf850ec87d5327a965db9-Abstract-Conference.html.

**Questions:**

Based on the concerns summarized in the weaknesses section, I would like to pose the following questions, categorized into “theoretical” and “experimental” aspects:

### Questions on Theoretical Analyses:
1.	Could the authors provide a more detailed discussion on the behavior of each term in the derived bound and how the bound behaves as the iteration $N$ approaches infinity, particularly in terms of convergence and conditions for convergence?
2.	In the convex case, variance reduction is guaranteed only near the optimal solution (Thm. 1), while the convergence analysis in Thm. 3 assumes variance reduction is achieved “in all iterations.” Does this assumption imply that parameters are initially close to the optimal solution, and if so, how realistic is this assumption?
3.	The improvement in the bound relies on the assumption that variance reduction is guaranteed for all iterations. How does this assumption align with the results of Thm. 1? Is the radius $r$ of the region always small enough to ensure consistency between the two theorems? If not, there may be a significant gap between these results.

### Questions on Experimental Analyses:
1.	The numerical experiments are primarily conducted on synthetic data. Could the authors consider conducting experiments on real-world datasets to provide more practical insights into the usefulness of the proposed method?
2.	There are no experiments measuring the reduction in gradient variance. Could the authors include empirical measurements of gradient variance to illustrate how much variance is reduced compared to existing methods? The following references may be relevant for this consideration: [Miller et al. (2017); Buckholz et al. (2018)].
3.	In the proposed method, the degree of variance reduction appears to depend on the parameter $c$. Could the authors provide additional practical discussions and sensitivity analyses on the selection of $c$ and its impact on variance reduction performance, particularly in terms of gradient variance? Although Figure 2 presents the relationship between $c$ and the degree of variance reduction, it would be beneficial to confirm whether this behavior is consistent across various experimental settings.
4.	The comparison with the method of Roeder et al. (2017) in Euclidean SVI is understandable; however, could the authors clarify the rationale for not including comparisons with more recent variance-reduction methods?

**Details Of Ethics Concerns:**

N/A.

---

> ### Author Response · Authors · 2024-11-20
>
> Thank you for the detailed comments and the effort put into the review.
>
> # Concerns Regarding Theoretical Analyses:
>
> - Our convergence analysis is in two stages for clarity; We first show our estimator has a smaller variance than the Monte Carlo estimator in several scenarios, and then show that the benefit of variance reduction will propagate through the proof of SGVI [Diao2023] to improve their bounds. We can discuss the behaviour of the bounds as $N$ approaches infinity as follows. As $N \to \infty$, the term associated with $W_2^2(\mu_0,\hat{\pi})$ goes to zero (sublinearly for the convex case and linearly for the strongly convex case), while the constant term associated with the noise behaves like $O(\eta)$ where $\eta$ is the stepsize. For a fixed $\eta>0$, the constant terms specify a "convergence neighbourhood" where further convergence is not possible, especially due to the noise coming from the Hessian $\nabla^2 V(X_k)$ that does not diminish. However, our convergence neighbourhood is smaller than the one in SGVI of Diao et al., meaning that the variance reduction helps our algorithm reach closer to the optimal solution than SGVI.
>
> - First, we clarify that Theorem 1 holds also without the convexity assumption. For Theorem 3, the analysis can easily be relaxed to the later iterations when we are within the vicinity of the optimal solution (see our comment in lines 389 -- 391), similar to warm-up optimization: initially run the vanilla version of SGVI (simply set $c=0$) and switch to using SVRGVI (set, e.g., $c=0.9$) at the convergence phase. We also note that the convergence phase is where variance reduction is needed the most for stable convergence. The method always helps (in theory) for stable convergence around the optimum, and, in practice, provides benefits right from the start. We further note that for strongly log-concave targets, variance reduction happens also further from the optimal solution (Theorem 2).
>
> - For the smooth case the radius $r$ depends on the smoothness parameter $\ell$ of the Laplacian of the log target as well as the variance of the optimal distribution $\hat{\pi}$. The smoother the log-target and the larger the variance of the optimal distribution, the larger the radius. However, we currently do not have means of quantifying it or determining what is sufficiently large (and relative to what).
>
> # Concerns Regarding Numerical Experiments:
>
> - We build on SGVI and BWGD, both of which only considered synthetic data as well. We fully agree there is clear value in real-world examples, but for a still developing family of algorithms, the effects can be best demonstrated and quantified in controlled experiments. Note that we considered more targets and in higher dimensions, compared to the previous works.
>
> - This is a very good point. We now added new result figures (Appendix B.4) showing the variance reduction along iterations in practice -- we obtain substantially smaller variance compared to the standard single-sample Monte-Carlo estimator. This explains the variance reduction as the driving force for the improved convergence. Regarding $c$, we have added the sensitivity experiment in Appendix B.6, showing that $c$ being around $1$ works best in practice.
>
> - Our primary objective is to improve Bures-Wasserstein Variational Inference (BW-VI) methods and explain when and how they work. Note that we did not claim those methods (including ours) would beat all Euclidean Variational Inference (EVI) methods. Instead, the comparison against EVI was included to show that in some cases the previous BW-VI methods can be worse than rather standard Euclidean VI methods -- an important observation that is not apparent based on the previous literature -- and that this is at least partly due to the high variance of the gradient estimators that can be easily resolved. We chose the method of Roeder et al. (2017) as a representative reference method because it is also a variance-reduction method using a score and shares similar characteristics to our variance-reduction method in the Bures-Wasserstein space. In the revised version we will discuss newer EVI methods and point out thorough empirical comparison as a valuable future work.
>
>
> **Citations:** [Diao2023] Diao, M. Z., Balasubramanian, K., Chewi, S., \& Salim, A. Forward-backward Gaussian variational inference via JKO in the Bures-Wasserstein space. ICML 2023.
>
> # Lack of Discussion on Other Variance Reduction Studies in Euclidean VI:
>
> More advanced techniques beyond control variates in Euclidean VI are very appreciated and it would be a promising future work to integrate these techniques to further improve BW VI. We extended the discussion on Euclidean VI as suggested in lines 503-506. However, since we have not yet studied these methods in our setting, we are not able to discuss what challenges they may cause.

---

> > ### Author Response · Authors · 2024-11-20
> > **Rebuttal (continued)**
> >
> > About the questions: most questions are also already addressed in our previous comment on the "concerns". We include again here for completeness.
> >
> >
> > # Questions on Theoretical Analyses
> >
> > 1. Yes, we answered this question in the first bullet point of "concerns regarding theoretical analyses". We will incorporate that discussion in the final version.
> >
> > 2. As answered above, we can relax this assumption for later iterations only. We also note that the later iterations are where variance reduction is needed the most for stable convergence.
> >
> > 3. About the radius $r$, as answered above, we do not really know how large it is. However, we can say something about it like: "The smoother the log-target and the larger the variance of the optimal distribution, the larger the radius."
> >
> > # Questions on Experimental Analyses
> >
> > 1. As answered to Reviewer 7JyP, we leave empirical comparisons on more challenging tasks as future work; they need more work on the careful tuning of learning rates, more robust implementations, etc. that could not be fit in this paper.
> >
> > 2. Yes, as answered above, we now measured the variance along the iterates (Appendix B.4), showing that our method substantially and consistently reduces the variance.
> >
> > 3. Yes, as answered above, we provided the sensitivity discussion on $c$ in Appendix B.6.
> >
> > 4. Yes, we answered this question above.

---

> ### Comment · Reviewer_qVPq · 2024-11-22
> **Reply for rebuttal**
>
> Thank you for your detailed and thoughtful rebuttal. While your responses have addressed several of my concerns, I still have some remaining questions and would appreciate further clarification on the following points:
>
> ### Additional Questions for Theory:
> - Applicability of Theorem 2 to Convex Functions:
>   - Based on Theorem 2, variance reduction is claimed to be achievable for all iterations. However, this result seems to hold only under the $\alpha$-strongly convex assumption. Does this imply that Theorem 2 cannot be directly applied to the general convex case covered in Theorem 3? If so, could Theorem 2 be extended to address the general convex scenario?
> - Condition $\mathrm{Tr}(\Sigma^{-1}) < \frac{2\alpha d}{c}$ in Theorem 2:
>   - Is this condition generally satisfied in practical scenarios? Remark 2 suggests that achieving variance reduction requires aligning the target distribution properties with $\alpha$ and calculating $\lambda_{\mathrm{min}}(\Sigma)$ at each iteration. However, this step is not included in Algorithm 1. How should practitioners use the proposed method for approximating target distributions based on convex functions in such cases?
> - Practical Intuition for Theorem Conditions:
>   - The assumptions and results in theorems, such as Theorem 2’s variance reduction or Theorem 1’s smaller variance, are not intuitively clear in terms of practical implications (this reflect my lack of understanding, maybe?). For instance, according to the rebuttal, variance reduction in Theorem 1 depends on $r$, but identifying $r$ is challenging, leaving it unclear whether the reduced variance is sufficiently small. Does this property align with the goal of ensuring small variance at each iteration? How does the proposed method compare to variance reduction approaches based on sampling techniques, such as Quasi-Monte Carlo?
>
> ### Additional Questions for Experiments
> - Importance of Hyperparameter Settings:
>   - As indicated by Theorem 3, settings such as learning rates are crucial for achieving the theoretical results. Would the authors agree that further experimentation and discussions on this aspect are necessary prior to publication?
> - Visualization and Sensitivity Analyses:
>   - Could the authors use an anonymous image-sharing service to share results showing measured variance and sensitivity analyses? These results are critical for verifying whether the claimed variance reduction is genuinely achieved by the proposed method.
>
> - Regarding the comparison to EVI with variance reduction
>   - I believe the current demonstration of the advantages of the proposed method over EVI-based approaches still appears insufficient. For example, Figure 3 shows that EVI with variance reduction outperforms both SGVI and BWGD. This raises the concern that more advanced variance reduction techniques developed for EVI might surpass the proposed method in performance.
> If feasible, could you provide comparisons with recent state-of-the-art variance reduction techniques in EVI to clarify how the proposed method stands in relation to them?

---

> ### Author Response · Authors · 2024-11-24
> **Author rebuttal -- round 2**
>
> We thank the reviewer for the reply and the thoughtful comments.
>
> **First, we address that we updated the paper, so please check out our new PDF file in OpenReview.**
>
> # Additional Questions for Theory:
>
> It is more natural for us to address your three bullet points in the following order: second - first - third.
>
>
> - **Condition $Tr(\Sigma^{-1})<\frac{2\alpha d}{c}$:** Yes, we can verify and check this condition in practical scenarios. Theorem 2 says that: for $\alpha$-strongly convex $V$,  we obtain variance reduction whenever $c<(2\alpha d)/Tr(\Sigma^{-1})$. We also note that, at the current iteration, we already have the Cholesky decomposition of $\Sigma$ as $\Sigma = L L^{\top}$, available from the sampling step. Therefore, we can compute $Tr(\Sigma^{-1})$ based on the triangular matrix $L$ with the complexity of $O(d^2)$, which is neglectable. Therefore, we can always guarantee variance reduction if we, at the current iteration, pick $c$ smaller than the computable term $(2\alpha d)/Tr(\Sigma^{-1})$. This is also our recommendation for the practitioner in the worst case -- if they want to be safe; However, in practice, $c$ can be chosen more flexibly. For example, as stated in Theorem 1, in a neighbourhood of the optimal solution, any $c \in (0,2)$ should guarantee variance reduction (even without requiring the target to be log-concave!).
>
>
> - **Applicability of Theorem 2 to Convex Functions:** Yes, we can extend to convex case ($\alpha=0$). We assume that $V$ is twice continuously differentiable. We show that: for any Gaussian random variable $X$, $Tr(\mathbb{E} \nabla^2 V(X)) > 0$. By contradiction, suppose that $Tr(\mathbb{E} \nabla^2 V(X)) = 0$, it follows that $\mathbb{E} \nabla^2 V(X) = 0$ since $\mathbb{E} \nabla^2 V(X)$ is symmetric and positive semidefinite. Therefore, for all $z$: $z^{\top} (\mathbb{E} \nabla^2 V(X)) z = 0$ or $\mathbb{E}(z^{\top} \nabla^2 V(X) z) = 0$. Let's denote by $f$ the pdf of $X$, we have: given any fixed $z$:
> \begin{align*}
>     \int{(z^{\top} \nabla^2 V(x) z) f(x)dx} = 0
> \end{align*}
> Since $V$ is convex, the function under the integral is non-negative. Therefore, the integral being zero implies that the function is zero almost everywhere. The continuity of $\nabla^2 V$ further implies that the function under the integral has to be identically zero. Since $f(x)>0$ for all $x$, we deduce $z^{\top} \nabla^2 V(x) z = 0$ for all $x$. Now pick $z = e_i$ where $e_i$ is the i-th basis vector, i.e., the i-th component is $1$, and $0$ otherwise,  we get $\partial^2_{x_i^2} V \equiv 0$. For $i \neq j$, we set $z$ so that $z_i = z_j = 1$, $z_k=0$ for all $k \neq i,j$, we have $\partial^2_{x_i^2} V + \partial^2_{x_j^2} V + 2 \partial^2_{x_i x_j} V \equiv 0$, or $\partial^2_{x_i x_j} V \equiv 0$. We conclude that $\nabla^2 V \equiv 0$, or $V$ is an affine function. This cannot happen since $V = -\log(\pi)$. This contradiction implies that $Tr(\mathbb{E} \nabla^2 V(X)) > 0$. From Lemma 1, we deduce that variance reduction happens whenever $0< c< 2 Tr(\mathbb{E} \nabla^2 V(X)) / Tr(\Sigma^{-1})$. In this case, the upper bound of $c$ is not given as explicitly as in the strongly convex case, but the point here is that variance reduction still happens in this context.
>
>
> - **Practical Intuition for Theorem Conditions:** First, as answered above, the practical implication for Theorem 2 is now clear. We empirically verify in Appendix B.4 that we obtained strong variance reduction effectively from the beginning. About the comparison to the trivial approach such as Monte Carlo with multiple samples (see also our answer to **Reviewer NBYM**), we added an experiment in Appendix B.5, showing that the Monte Carlo method needs roughly **100 samples** (hence gradient oracles) per iteration to reach our performance using just **one sample** per iteration.
>
> # Additional Questions for Experiments
>
> - **Importance of Hyperparameter Settings:** Yes, we now added a study on the stepsize in Appendix B.7, showing that our method strongly outperforms previous BW methods under different stepsizes, our method is also less sensitive and allows using larger step sizes.
>
> - **Visualization and Sensitivity Analyses:** Yes, we measured the variance along iterations in Appendix B.4 and performed sensitivity analyses regarding $c$ in Appendix B.6.
>
> - **Regarding the comparison to EVI with variance reduction:** we are trying to incorporate one more EVI as a baseline if time allows.

---

> ### Author Response · Authors · 2024-11-26
>
> Given the above rebuttal, we would like to additionally make this comment to resolve the **main concern** of the reviewer which is about the condition that "variance reduction holds for all iterations":
>
> In the strongly convex case (Theorem 4), **yes** we can guarantee variance reduction for all iterations if we set $c_k$ such that $0< c_k < 2\alpha d / Tr(\Sigma_k^{-1})$. In the convex (but not strongly convex) case  (Theorem 3), variance reduction also happens at all iterations if we set $c_k$ small enough.

---

> ### Author Response · Authors · 2024-11-27
> **Regarding EVI -- author rebuttal continues**
>
> In response to the additional experiment regarding EVI, we remark that Miller et al. (2017) consider only Gaussian with **diagonal** covariance, it is misleading and besides the point to compare it against our method that considers full covariance matrices. About Buckholz et al. (2018), the core idea of the method is to replace Monte Carlo samples with quasi-Monte Carlo samples to improve the estimator. While their idea was used in EVI, it is universal and it is straightforward to incorporate the idea into Bures-Wasserstein VI methods. Consequently, instead of plotting the results of Buckholz et al. (2018) directly, we adopted their quasi-Monte Carlo sampling approach to improve SGVI, enabling a more direct and meaningful comparison to our method. To this end, we have added a new experiment in Appendix B.5, which demonstrates that using quasi-Monte Carlo samples outperforms traditional Monte Carlo samples. However, to achieve comparable performance to our method SVRGVI (with one MC sample per iteration), SGVI requires around 50 quasi-Monte Carlo samples per iteration. With 100 quasi-Monte Carlo samples, SGVI surpasses our method (which still uses one sample per iteration).
>
> ## Citations:
>
> - Miller et al. (2017): Andrew C. Miller, Nicholas J. Foti, Alexander D'Amour, and Ryan P. Adams. Reducing Reparameterization Gradient Variance. NeurIPS2017. https://arxiv.org/abs/1705.07880
>
> - Buckholz et al. (2018): A. Buchholz, F. Wenzel, and S. Mandt. Quasi-Monte Carlo Variational Inference. ICML2018. https://arxiv.org/abs/1807.01604.

---

> > ### Comment · Reviewer_qVPq · 2024-11-30
> > **Reply from reviewer**
> >
> > I would like to express my gratitude and respect for the authors’ sincere efforts in engaging in the discussion, extending the theory, and providing additional experiments.
> > After reviewing the above discussions and the experimental results provided, I believe the contributions are sufficiently significant.
> >
> > I recommend reflecting the points raised during our discussion in the revised version, including the need for additional computations in hyperparameter tuning and the limitation that the theoretical guarantees do not hold for non-convex $V$ , as future work or known limitations.
> >
> > Therefore, I am changing my score from 3 to 6.
> > Best wishes for the success of your paper.
> >
> > Sincerely,
> > --Reviewer qVPq

---

> > > ### Author Response · Authors · 2024-11-30
> > >
> > > Thank you for your kind consideration of raising the score. We will include those points in the revised version.

---

### Official Review · Reviewer_Sicz · 2024-11-03

**Soundness:** 4
**Presentation:** 4
**Contribution:** 3
**Rating:** 8
**Confidence:** 4

**Summary:**

The paper proposes an improved optimization technique for variational inference with a multivariate Gaussian variational family.

Building on the prior work that proposed optimization on the Bures–Wasserstein manifold and simple one-sample stochastic gradient estimators, this work offers improved stochastic gradient estimators. These are based on a simple idea: as iterates get closer to the optimum, they provide additional information about the next gradient that can be used to reduce variance of its stochastic estimator.

Authors support the superiority of the new optimization technique with both theoretical analysis and empirical evaluation, the latter on a set of synthetic benchmarks.

**Strengths:**

- The paper is written very well. It is a pleasure to read.
- The theoretical analysis is convincing (I didn’t rigorously check the proofs but I don’t doubt the claims and I am pretty sure the methods authors use are quite appropriate).
- Empirical evaluation shows very strong improvement over the baselines, more so than the theoretical analysis (which is not claimed to be tight) suggests.

**Weaknesses:**

-I’m not an expert in optimization, so it’s somewhat hard for me to judge the impact. I can speculate that the problem might seem somewhat narrow: variational inference with _Gaussian_ variational family. However, I don’t really think so myself. Furthermore, in my opinion, a good solution to even a somewhat narrow problem definitely warrants publication.

I would actually give this paper a score of 9, but only 8 and 10 are available.

**Questions:**

As mentioned above, in my opinion the paper is very well-written. I therefore don’t have any content-related questions, only a small number of typographical suggestions for the authors:
- Perhaps use the en-dash (--) instead of the simple dash (-) in "Bures–Wasserstein". I believe this is the standard when it comes to joining two names.
- I believe that "FB" abbreviation (forward-backward) is never introduced.
- The term “L-smooth” seemed weird to me, why not “L-Lipschitz”?
- Page 4, line 174, “affine map” -> “affine maps”.
- Page 4, line 198, “shows that” -> “showed that”.
- Page 5, line 233, I would remove parentheses around the “also see ...”, they look ugly immediately after a citep.
- Page 5, line 237, perhaps add commas around “where $c \in \mathbb{R}$”, otherwise the sentence is hard to read.
- Perhaps, when introducing Algorithm 1, you could briefly mention there are recipes for defining the constants $c_k$ that you investigate further in the text. An algorithm without parameters you don’t know how to set would come across as more useful.
- Page 8, line 387, “We also note that conditioning” -> “We also note that by conditioning”.
- Page 8, footnote 1, please expand what is the minor correction. Also, perhaps start with a capital letter.
- Page 9, line 457, missing “e” in “covariancs”.
- Page 9, line 472, “In The” -> “In the”.

Note: no need to respond to the small items above in the rebuttal, save yourself some time.

---

> ### Author Response · Authors · 2024-11-20
>
> Thank you for the thoughtful and positive comments on our work and the acceptance recommendation. We have addressed the minor points raised in the question section in the revised version.

---

> > ### Comment · Reviewer_Sicz · 2024-11-22
> >
> > I acknowledge reading the authors' rebuttal.

---

### Meta-Review · Area_Chair_Bdoa · 2024-12-18

**Metareview:**

The proposed work claims that recent variational inference methods based on stochastic implementations of the Bures-Wasserstein gradient flow can be improved by an order-of-magnitude. This is achieved by the principle of control variates.

The strength of the work is the theoretical analysis which gives an improved convergence bound for BW variational inference. The simplicity of the presented method and its good numerical performance is another strength. All reviewers found the paper well written and appreciated the overall presentation.

The main weaknesses are the lack of larger scale numerical experiments, and lack of comparison and acknowledgment of existing works on control variates in VI or even simpler techniques like momentum. Moreover, there are works on variational inference in different geometries (Fisher-Rao) which are arguably more natural for Bayesian inference than the Wasserstein geometry (which is close to the Euclidean geometry). These weaknesses were mostly addressed by the authors in the rebuttal phase.

Overall I recommend acceptance of the work. The simplicity of the approach, its promising numerical performance as well as the theoretical results will be of interest to the community. I recommend the authors to take the feedback of the reviewers into account for the final version of the manuscript.

**Additional Comments On Reviewer Discussion:**

Reviewer qVPq initially raised several issues regarding the theoretical results, numerical evaluation and relation to other control variate methods. These were mostly cleared up in the rebuttal phase, and the score was raised from 3 to 6.

All reviewers liked the simplicity of the approach, the promising numerical results and clarity of writing, which leads me to recommend acceptance.

---

### Decision · Program_Chairs · 2025-01-22

Accept (Poster)